# Human immunoglobulin G hinge regulates agonistic anti-CD40 immunostimulatory and antitumour activities through biophysical flexibility

Xiaobo Liu [1,2,7], Yingjie Zhao[1,2,7], Huan Shi[1,2,7], Yan Zhang[1,2], Xueying Yin [3], Mingdong Liu, Huihui Zhang, Yongning He[4], Boxun Lu [5], Tengchuan Jin [3] & Fubin Li [1,2,6]

Human immunoglobulin G (IgG) agonistic antibodies targeting costimulatory immunoreceptors represent promising cancer immunotherapies yet to be developed. Whether, and how, human IgG hinge and Fc impact on their agonistic functions have been disputed. Here, we show that different natural human IgGs confer divergent agonistic anti-CD40 immunostimulatory and antitumour activities in FcγR-humanized mice, including inactive IgG3 and superior IgG2. This divergence is primarily due to their CH1-hinges despite all human IgGs requiring Fc-FcγR binding for optimal agonistic activities. Unexpectedly, biophysical flexibility of these CH1-hinges inversely correlates with, and can modulate, their agonistic potency. Furthermore, IgG Fcs optimized for selective FcγR binding synergize with and still require IgG hinge, selected for rigidity, to confer improved anti-CD40 immunostimulatory and antitumour activities. These findings highlight the importance of both hinge rigidity and selective FcγR binding in antibody agonistic function, and the need for newer strategies to modulate antibody agonism for improved clinical application.

[1] Shanghai Institute of Immunology, Faculty of Basic Medicine, Shanghai Jiao Tong University School of Medicine, Shanghai 200025, China. [2] Key Laboratory of Cell Differentiation and Apoptosis of Chinese Ministry of Education, Shanghai Jiao Tong University School of Medicine, Shanghai 200025, China. [3] Laboratory of Structural Immunology, CAS Key Laboratory of Innate Immunity and Chronic Disease, CAS Center for Excellence in Molecular Cell Science, School of Life Sciences and Medical Center, University of Science and Technology of China, Hefei, Anhui 230027, China. [4] State Key Laboratory of Molecular Biology, National Center for Protein Science, Shanghai Science Research Center, Institute of Biochemistry and Cell Biology, Shanghai Institutes for Biological Sciences, Chinese Academy of Sciences, Shanghai 201210, China. [5] State Key Laboratory of Genetic Engineering, Department of Biophysics, School of Life Sciences, Fudan University, Shanghai, China. [6] Collaborative Innovation Center of Systems Biomedicine, Shanghai Jiao Tong University, Shanghai, China. [7] These authors contributed equally: Xiaobo Liu, Yingjie Zhao, Huan Shi. Correspondence and requests for materials should be addressed to F.L. (email: fubin.li@sjtu.edu.cn)

In order to target and mobilize immune cells to eradicate tumours, two classes of antibodies have been pursued[1,2]: (1) blocking antibodies against immunoinhibitory receptors or ligands and (2) agonistic antibodies for immunostimulatory receptors. Clinically approved antibodies targeting CTLA-4 and PD-1/PD-L1 all belong to the first category[3]. While these antibodies prove the concept of cancer immunotherapy and represent major breakthroughs, their limitations, including rather low overall response rates, highlight an urgent need to develop additional therapies with distinct mechanisms of action[3,4]. Agonistic antibodies targeting immune stimulatory molecules have been suggested as a class of promising therapeutic candidates in both animal and clinical studies[5]. It is, therefore, critical to understand how antibody agonistic function is regulated.

CD40 is a tumour necrosis factor receptor (TNFR) superfamily member expressed broadly on antigen-presenting cells (APCs), and controls a key immunostimulatory pathway required for humoral and cellular immunity[6,7]. Ligation of CD40 by agonistic anti-CD40 antibodies, surrogates of CD40L expressed by CD4 T cells, can effectively promote APC maturation and in turn antigen-specific CD8[+] cytotoxic T-cell activation and expansion, and strengthen antitumour responses[6–8]. Agonistic anti-CD40 antibodies have also been reported to stimulate tumour-infiltrating macrophages that contribute to tumour eradication by depleting tumour stroma[9]. Using murine agonistic antibodies targeting CD40, we and others have previously demonstrated that IgG constant domains (CDs) can dictate their immunostimulatory and antitumour activities by binding to different Fcγ receptors (FcγRs)[10,11]. Both mice and humans express several activating (FcγRI, FcγRIII, and FcγRIV in mice and FcγRI, FcγRIIa, and FcγRIIIa in humans) and one inhibitory (FcγRIIB) type-I FcγRs, classified based on their associated intracellular signalling motifs[12]. Murine agonistic anti-mouse CD40 antibodies were found to specifically depend on Fc–FcγRIIB interactions, in contrast with effector antibodies that require activating FcγRs[10,11]. Furthermore, murine agonistic IgG antibodies targeting several other TNFR superfamily members, including DR5, FAS, and CD27, have also been shown to be regulated by their CDs in a similar way[13,14], suggesting that murine IgG agonism might be generally regulated by their CDs through differential FcγR binding. Mechanistically, we and others have suggested that murine IgG Fc's can contribute to agonism by promoting FcγR-dependent in vivo cross-linking and therefore clustering of the targeted receptors such as CD40, mediated by FcγR-expressing cells that work in trans[11,15].

The regulation of the agonistic function of human IgG (hIgG) antibodies by their CDs appears to be more complicated and disputed. HIgG has four subclasses, namely IgG1–4, with similar structure but distinct effector function and implication in immune responses and diseases[16]. In addition to different FcγR-binding properties, hIgGs also have different hinge regions with different biophysical flexibility, which has been reported to positively correlate with antibody effector functions[17–19]. Previously we showed that both human IgG1 and 2 anti-CD40 antibodies depend on their FcγR-binding ability for agonistic activities, and that Fc variants with selectively enhanced binding to FcγRIIB have significantly improved agonistic potency[10,20]. In contrast, White et al. have reported that, under the mouse FcγR background, human IgG2 agonism does not depend on Fc–FcγR interactions, but rather on its unique hinge conformation[21]. However, it appears that the reported impact of IgG2 hinge conformation does not apply to all anti-CD40 antibodies[22], nor does it apply in FcγR-humanized mice[20]. Therefore, the impact of hIgG hinge and Fc on its agonistic function, under physiological conditions with human FcγRs expression, remains to be elucidated.

In order to address this issue, we evaluated, in an FcγR-humanized mouse model, the agonistic function of multiple anti-CD40 and -DR5 antibody sets with different natural and engineered hIgG Fc's and CH1-hinges. We also characterized the biophysical flexibility of IgG hinges to investigate their mode of action in modulating antibody agonistic function. Here, we show that different natural hIgGs confer divergent agonistic potency in multiple anti-CD40 antibodies, and that IgG hinge and Fc domains can regulate antibody agonism through distinct mechanisms, based on which agonistic anti-CD40 antibodies with improved immunostimulatory and antitumour activities are developed.

## Results

**Natural hIgG CDs confer divergent agonism**. To investigate the impact of hIgG CDs on antibody agonistic function, we generated anti-mouse CD40 antibodies with different natural hIgG CDs. These antibodies exhibited similar specificity and affinity for mouse CD40 (mCD40), as shown by their competition kinetics against the parental rat anti-mCD40 antibody 1C10 (Supplementary Fig. 1a, Supplementary Table 1) and similar binding kinetics to mCD40 (Supplementary Fig. 1b). Agonistic activities of these antibodies were evaluated in a physiologically relevant mouse model that recapitulates the expression profile of human FcγRs[23], referred to as "FcγR-humanized mice." Agonism of anti-CD40 antibodies was evaluated by their immunostimulatory activity in an OVA-specific CD8[+] T-cell response model[10,15], where OVA antigen is delivered to dendritic cells (DCs) in the form of OVA/anti-DEC205 fusion protein (DEC-OVA), and agonistic anti-CD40 antibodies can promote maturation of DCs, activation and expansion of transferred OVA-specific OT-I CD8[+] T cells in recipient mice (Fig. 1a, b).

When anti-CD40 antibodies with different natural hIgG CDs were evaluated in this mouse model, mice treated with human IgG1, 2, and 4 anti-mCD40 antibodies displayed increased percentage of OT-I cells among CD8[+] T cells (Fig. 1b, c) and percentage of IFN-γ[+] CD8[+] T cells (Supplementary Fig. 2a), as compared to mice treated with control antibodies. However, human IgG3 antibodies failed to show any significant activities at the same dosage of 30 μg per mouse. To further investigate the relative potency of these antibodies, we titrated antibody dosage down to 10 μg per mouse. At this dosage, human IgG2 antibodies displayed robust activity, whereas human IgG1, 3, and 4 antibodies all failed to induce significant OVA-specific CD8[+] T-cell response (Fig. 1d). Strikingly, ten times more human IgG3 anti-mCD40 antibodies (100 μg per mouse), did not show any significant agonistic activity neither (Supplementary Fig. 2b). These data demonstrated that four natural hIgG CDs confer very different levels of anti-mCD40 antibody agonism, with IgG3 being agonistically inactive and IgG2 being agonistically superior.

**Optimal hIgG in vivo agonism requires Fc–FcγR engagement**. To test whether the different FcγR-binding property of natural hIgG Fc's[24] is responsible for their varying ability in driving anti-mCD40 antibody agonism, we examined whether FcγR expression is required for the agonism of hIgG anti-mCD40 antibodies using FcγR-deficient mice[23]. As shown in Fig. 1b, c, and Supplementary Fig. 2a, none of human IgG1–4 anti-mCD40 antibodies showed significant immunostimulatory activities in FcγR-deficient mice. Consistently, we also observed that the human IgG2 anti-mCD40 antibody variant carrying the N297A mutation, which abrogates Fc–FcγR binding[10], has no agonistic activity (Fig. 1e). Therefore, Fc–FcγR engagement is required for optimal in vivo agonistic potency of all hIgG isotypes.

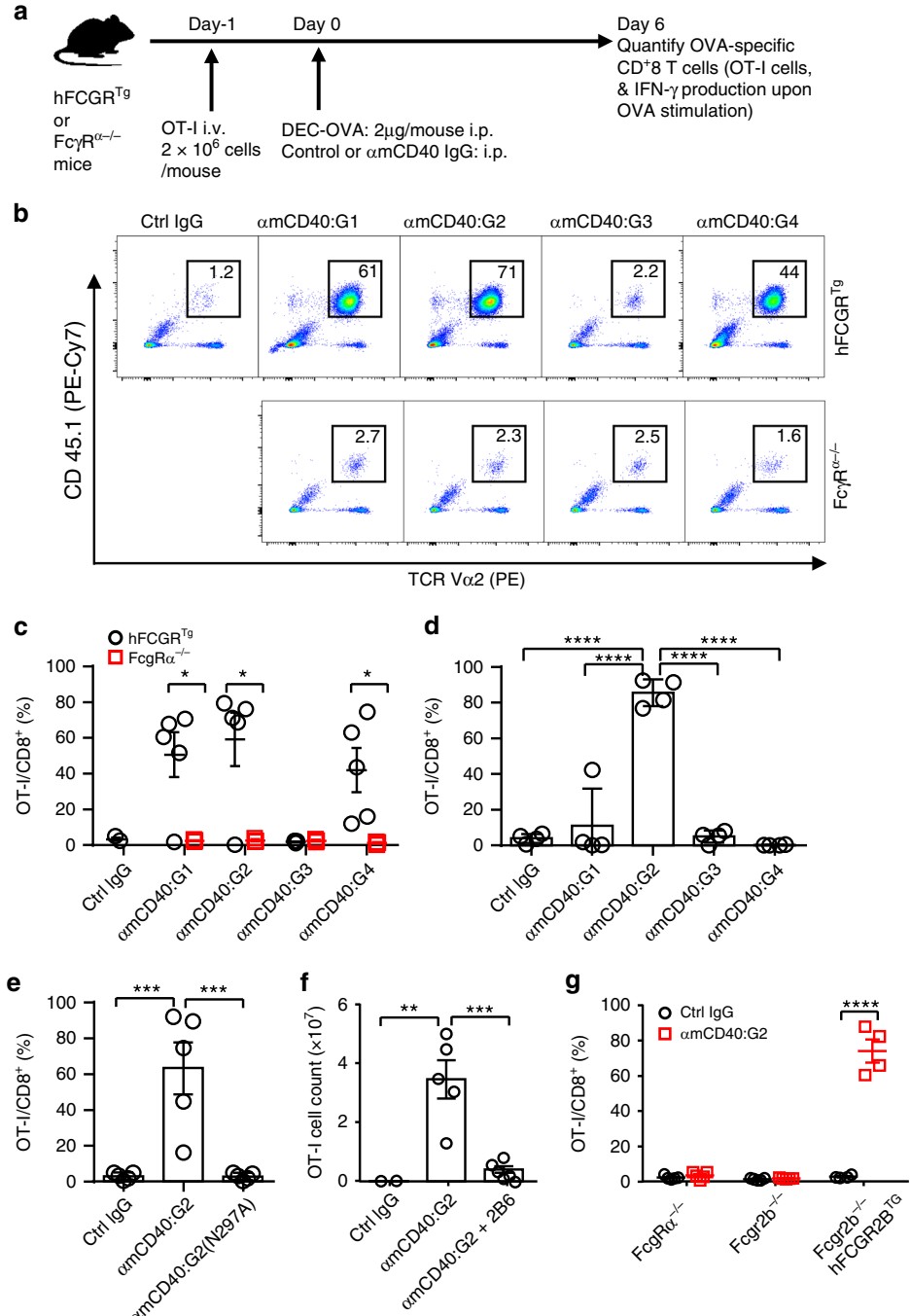

**Fig. 1** Divergent hIgG agonism and specific FcγR-binding requirement. **a** Diagram showing the OVA-specific CD8+ T-cell response model. In brief, FcγR-humanized (hFCGR$^{Tg}$) or -deficient (FcγRα$^{-/-}$) mice were adoptively transferred with OT-I cells on day-1, immunized intraperitoneally (i.p.) with 2 μg of DEC-OVA in the presence of control or anti-CD40 antibodies on day 0. Splenocytes were harvested to quantify OVA-specific CD8+ T cells on day 6. **b**, **c** Representative FACS profile (**b**) and quantification (**c**) showing the percentage of OT-I cells (CD45.1+TCRVα2+) among CD8+ T cells in mice treated and analyzed as in (**a**) together with 30 μg of indicated control or anti-mCD40 antibodies. Numbers of mice: **b**, **c** three hFCGR$^{Tg}$ mice for Ctrl IgG, five hFCGR$^{Tg}$, and three FcγRα$^{-/-}$ mice for other groups. **d**-**f** Quantification of OT-I cells as the percentage of OT-I cells among CD8+ T cells (**d**, **e**) or cell count (**f**) in FcγR-humanized mice treated and analyzed as in (**a**) together with 10 μg (**d**, **e**) or 30 μg (**f**) of indicated control or anti-mCD40 antibodies (the N297A mutation abrogates Fc–FcγR binding) and with/without FcγRIIB-blocking antibody 2B6 (150 μg per mouse) (**f**). Numbers of mice: **d** four mice per group; **e** five mice per group; **f** two mice for Ctrl IgG, five mice for αmCD40:G2, six mice for αmCD40:G2 + 2B6. **g** Quantification of OT-I cells as the percentage of OT-I cells among CD8+ T cells in mice of indicated genotypes (FcγR-deficient (Fcgrα$^{-/-}$, five mice per group), FcγRIIB-deficient (Fcgr2b$^{-/-}$, five mice per group) or humanized (Fcgr2b$^{-/-}$hFCGR2B$^{Tg}$, four mice per group)) treated and analyzed as in (**a**) together with 10 μg of indicated control or IgG2 anti-mCD40 antibodies. Each symbol represents an individual mouse. Bars represent the mean ± SEM. $^{*}p \leq 0.05$, $^{**}p \leq 0.01$, $^{***}p \leq 0.001$, $^{****}p \leq 0.0001$; unpaired two-tailed $t$ test (**c**, **g**), one-way ANOVA with Holm–Sidak's post hoc (**d**–**f**). Source data (**c**–**g**) are provided as a Source Data file. A representative of two independent experiments is shown

Since previous studies have shown that FcγRIIB-engagement is specifically required by murine agonistic anti-mCD40 antibodies for their activities[10,11], we investigated whether human FcγRIIB is important for the activities of human IgG2 anti-mCD40 antibodies. As shown in Fig. 1f, human FcγRIIB-blocking antibody 2B6 can significantly inhibit the activity of human IgG2 anti-mCD40 antibodies, suggesting that human FcγRIIB-engagement is necessary for their optimal activities. We further tested whether human FcγRIIB is sufficient in driving the immunostimulatory activities of human IgG2 anti-mCD40 antibodies, using FcγRIIB-humanized mice. While these antibodies displayed no significant activity in control $Fcgr2b^{-/-}$ mice, robust immunostimulatory activities were observed in FcγRIIB-humanized mice (Fig. 1g). In fact, much stronger agonistic activities were observed for human IgG2 anti-mCD40 antibodies in FcγRIIB-humanized mice (and FcγR-humanized mice) as compared to WT mice, suggesting that optimal human IgG2 agonism requires the FcγRIIB-humanized background (Fig. 1d and Supplementary Fig. 2c). The binding between human IgG2 antibodies and FcγR2B, although undetectable in ELISA (Supplementary Fig. 3a) likely due to its low affinity[24], was confirmed by surface plasmon resonance (Supplementary Fig. 3b).

Together, our study provided unequivocal evidence that Fc–FcγR interactions are required for the optimal agonistic activities of anti-mCD40 antibodies with natural hIgG CDs, and that among different Fc–FcγR interactions, human FcγRIIB engagement is not only necessary but also sufficient to drive the optimal agonism of human IgG2 anti-mCD40 antibodies.

**CH1-hinge is the basis of hIgG's divergent agonism**. We further investigated the structural basis of natural hIgG CDs' intrinsic difference in driving anti-mCD40 antibody agonism. Although human IgG2 has been reported to confer FcγR-independent agonistic function under the mouse FcγR background due to its unique CH1-hinge[21,22], the structural basis of the agonistically inactive IgG3 is not clear, nor is that of the much stronger FcγR-dependent agonism of IgG2 in FcγR-humanized mice.

HIgG Fc's were first tested given their unique FcγR-binding profiles (ref. [24] and Supplementary Fig. 3a) and the Fc–FcγR engagement requirement for hIgG agonism. A series of anti-mCD40 antibodies with CH1-hinge/Fc chimeric IgG CDs were generated (Supplementary Table 2). These antibodies were confirmed to have comparable FcγR-binding properties as their parental IgG antibodies with matched Fc domains (Supplementary Fig. 3a), as well as comparable binding to mCD40 (Supplementary Fig. 1b). When combined with IgG2 CH1-hinge (H2), all natural hIgG Fc's supported clear anti-mCD40 agonistic activities at a very low dosage of 3.16 μg per mouse (Fig. 2a). Interestingly, IgG2 Fc displayed the least agonistic potency as compared to other IgG Fc's (Fig. 2a). These data suggest that Fc is neither the structural basis of the agonistically inactive IgG3 nor that of agonistically superior IgG2. These data also suggest that the superior agonism of IgG2 in FcγR-humanized mice is based on and can be transferred along with its CH1-hinge.

We further tested whether CH1-hinge is the basis of the agonistically inactive IgG3. G2(H3) with the IgG2-Fc and IgG3 CH1-hinge (H3) was analyzed. As shown in Fig. 2b, G2(H3) anti-mCD40 antibodies failed to show any significant agonistic activities, suggesting that CH1-hinge is also responsible for the agonistically inactive property of IgG3. We further tested the combination of H3 and "V11" Fc, one of the most agonistically potent IgG1 Fc variants with optimized FcγRIIB binding[20,25]. V11 Fc was confirmed to have enhanced binding to human FcγRIIB (Supplementary Fig. 3c) and confer greatly improved

agonistic potency when combined with IgG1 CH1-hinge (H1) in our anti-mCD40 antibodies (Supplementary Fig. 4a, b). Strikingly, when the V11 Fc was combined with IgG3 CH1-hinge in V11(H3), it failed to show any significant agonistic activity in anti-mCD40 antibodies (Fig. 2c), despite V11(H3) antibodies having comparable mCD40 and human FcγR-binding profiles as V11(H1) antibodies (Supplementary Figs. 1b and 3c). These results demonstrate that human IgG3 CH1-hinge deprives V11 Fc of its strong agonistic potency. Together, our data suggest that CH1-hinge region is the structural basis of both agonistically inactive IgG3 and superior IgG2 CDs, and that their divergent agonistic potencies can be transferred along with their different CH1-hinges.

To test whether the divergent agonistic potency of human IgG2 and IgG3 CDs based on their CH1-hinge regions also apply to anti-human CD40 (hCD40) antibodies, we constructed two sets of anti-hCD40 antibodies based on the published clones 21.4.1 and 3.1.1[26], with these hIgG CDs. These two clones were confirmed to bind to hCD40 and have different binding epitopes as they do not block each other for binding to hCD40 and have different ability to block human CD40–CD40L interaction (Supplementary Fig. 3d, Supplementary Table 1, and refs. [26,27]). When these antibodies were evaluated in the human CD40/FcγR-transgenic mice[20], 21.4.1 antibodies displayed stronger immunostimulatory activities as compared to matched 3.1.1 antibodies (Fig. 2d–f), likely due to their special binding epitope[22]. However, we consistently observed that human IgG3 CD supports significantly less or no immunostimulatory activities as compared to human IgG2 CD in both 21.4.1 (Fig. 2d, e) and 3.1.1 (Fig. 2d, f) antibodies. Importantly, we observed that the ability of both human IgG2 and three CDs in driving anti-hCD40 agonistic activities, can also be transferred along with their CH1-hinges (Fig. 2d–f). These data, together with our anti-mCD40 antibody data, demonstrate that while natural hIgG CDs require Fc–FcγR interactions to drive optimal anti-CD40 antibody agonism, the intrinsically different agonistic potency of these natural IgG CDs, including the inactive IgG3 and the superior IgG2 CDs, is generally due to their different CH1-hinge, not Fc or association with specific binding epitopes.

**CH1-hinge and Fc impact on anti-CD40 antitumour activities**. The impact of hIgG CH1-hinge and Fc domains on anti-CD40 antibody agonism was further studied in antitumour responses. As shown in Fig. 3a, b, human IgG2 anti-mCD40 antibodies significantly inhibited MC38 colon cancer cell growth in FcγR-humanized mice whereas IgG1 and IgG3 did not, consistent with our OVA-specific CD8+ T-cell response studies (Fig. 1b–d). Also consistently, the IgG2(N297A) variants failed to show significant antitumour activities in FcγR-humanized mice (Fig. 3a, b), nor did IgG2 in FcγR-deficient mice (Supplementary Fig. 5a), confirming that Fc–FcγR interaction is critical for the antitumour activities of IgG2 anti-mCD40 antibodies. Importantly, although the Fc-optimized V11(H1) anti-mCD40 antibodies significantly inhibited MC38 tumour growth in FcγR-humanized mice, V11 (H3) displayed no antitumour activity (Fig. 3a, b), demonstrating that IgG3 CH1-hinge can deprive the V11 antibodies of their antitumour activities.

Antitumour activities of these anti-mCD40 antibodies were also studied in the MO4 melanoma model, and similar results were obtained. Human IgG1 antibodies displayed marginal antitumour activities, IgG2 and V11(H1) antibodies displayed stronger antitumour activities (Fig. 3c). However, none of the antibodies with reduced FcγR binding or IgG3 CH1-hinge, including IgG2(N297A), IgG3 and V11(H3), displayed any significant antitumour activity. These studies suggest that the

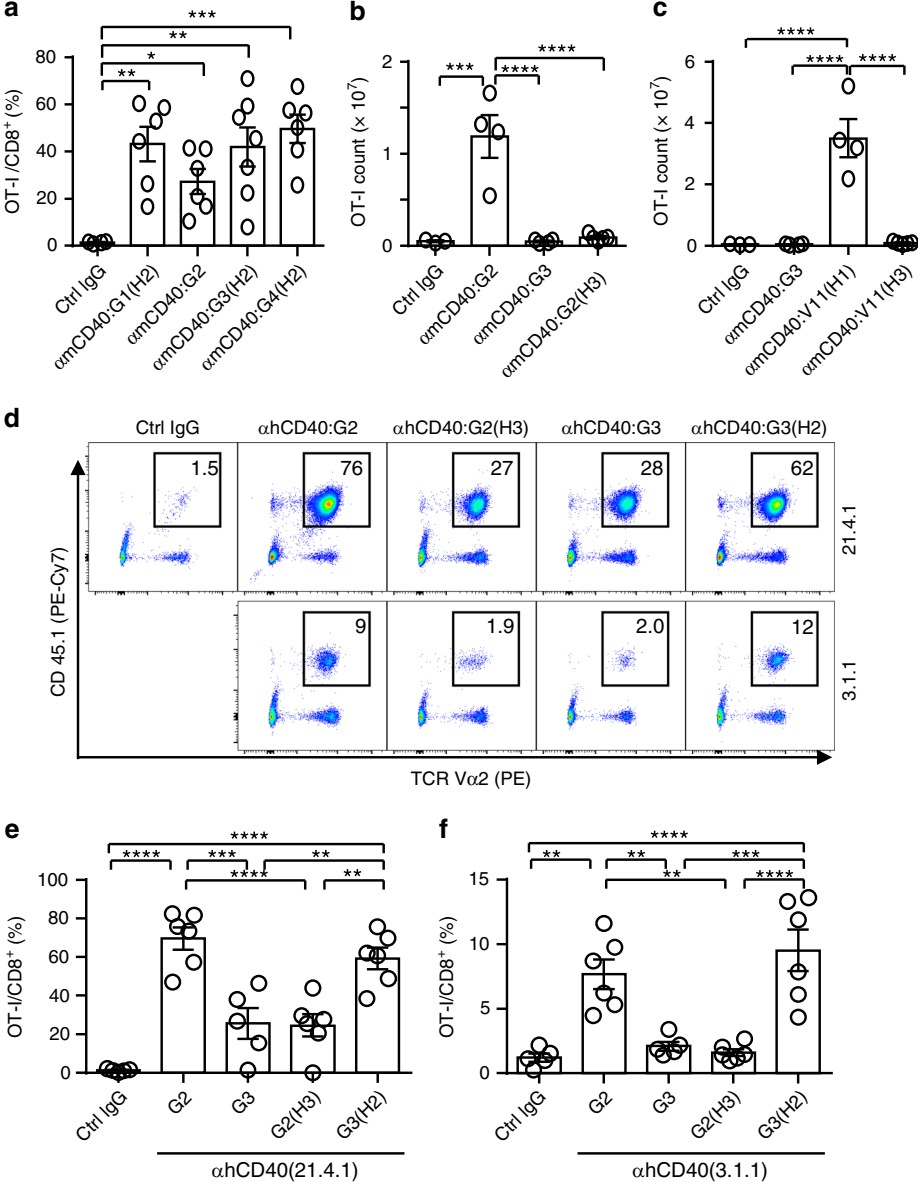

**Fig. 2** CH1-hinge basis of agonistically inactive IgG3 and superior IgG2. **a–c** Quantification of OT-I cells as the percentage of OT-I cells among CD8+ T cells (**a**) or cell count (**b, c**) in FcγR-humanized mice treated and analyzed as in Fig. 1a together with indicated control or anti-mCD40 antibodies (**a** 3.16 μg per mouse; **b, c** 10 μg per mouse). Numbers of mice: **a** six mice for Ctrl IgG, six to seven mice per group for others; **b, c** three mice for Ctrl IgG, four to five mice per group for others. **d–f** Representative FACS profile (**d**) and quantification (**e, f**) showing the percentage of OT-I cells (CD45.1+TCRVα2+) among CD8+ T cells in human CD40/FcγR-transgenic mice treated and analyzed as in Fig. 1a together with control or anti-human CD40 antibodies of indicated clones (Clones 21.4.1 and 3.1.1 have been described in Patent No.:US 7,338,660; Clone 21.4.1 in the IgG2 form is also known as CP-870,893) and constant domains (30 μg per mouse). Numbers of mice: **d–f** five to six mice per group. Each symbol represents an individual mouse. Bars represent the mean ± SEM. $^*p \leq 0.05$, $^{**}p \leq 0.01$, $^{***}p \leq 0.001$, $^{****}p \leq 0.0001$; one-way ANOVA with Holm–Sidak's post hoc. Source data (**a–c, e, f**) are provided as a Source Data file. A representative of two independent experiments is shown

impact of hIgG CH1-hinge and Fc domains on anti-CD40 antibody immunostimulatory activities can be translated into that on anti-CD40 antibody antitumour activities.

**The human IgG3 CH1-hinge confers the most flexibility.** Agonistic activities of anti-CD40 antibodies depend on their ability to hold CD40 molecules and promote their multimerization, which triggers CD40 downstream signalling[28]. Previously we and others have suggested that murine IgG Fc's can contribute to agonism by promoting FcγR-dependent in vivo cross-linking of anti-CD40 antibodies[11,15]. We hypothesized that different hIgG CH1-hinge regions have different biophysical

properties that could influence the ability of anti-CD40 antibodies to hold CD40 stably and to promote CD40 multimerization. Previous studies of hIgG using immunoelectron microscopy and fluorescence anisotropy have indicated that different hIgG hinges have different levels of flexibility[17,18]. Since these studies of antibodies require either non-physiological conditions or specific antigen-binding, we analyzed the flexibility of our anti-CD40 antibodies by small-angle X-ray scattering (SAXS) studies in their native forms[29].

Serially diluted monomeric anti-CD40 antibodies, purified and verified by gel-filtration (Supplementary Fig. 6a), were subjected to SAXS studies (Supplementary Table 3). The pattern of SAXS scattering intensity plots (Supplementary

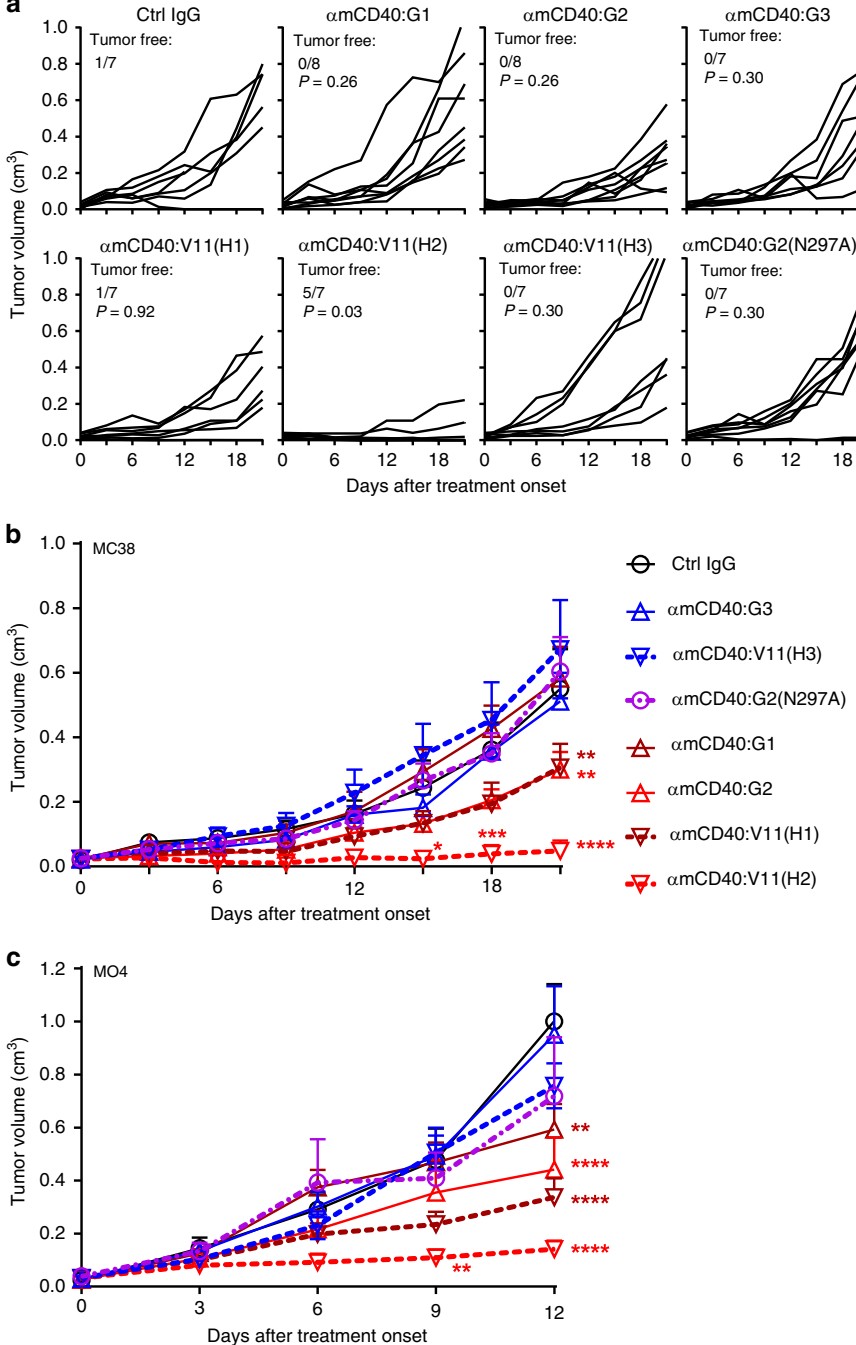

**Fig. 3** Impact of both IgG CH1-hinge and Fc on anti-CD40 antibody antitumour activities. MC38 (**a**, **b**) and MO4 (**c**) tumour volumes in FcγR-humanized mice following treatment with control or anti-mCD40 antibodies of indicated constant domains. After tumour cells were subcutaneously inoculated and established in FcγR-humanized mice, mice were treated i.p. twice on day 0 (the day when mice with palpable tumours receive their first treatment) and day 3 with 31.6 μg/mouse of control or anti-CD40 antibodies of indicated constant domains (the N297A mutation abrogates Fc–FcγR binding), and monitored for tumour growth. For mice inoculated with MO4 tumour cells in (**c**), each treatment also included 2 μg/mouse of DEC-OVA. Shown are tumour growth curves of individual mice (**a**) or mouse groups (**b**, **c**). Numbers of mice: **a**, **b** seven to eight mice per group; **c** seven mice per group except six mice for αmCD40:G2(N297A). Bars represent the mean ± SEM. $^{*}p ≤ 0.05$, $^{**}p ≤ 0.01$, $^{***}p ≤ 0.001$, $^{****}p ≤ 0.0001$, chi-square test (**a**), and two-way ANOVA with Holm–Sidak's post hoc (**b**, **c**) were used for group comparison. Source data (**a**–**c**) are provided as a Source Data file. A representative of two independent experiments is shown (**a**, **b**)

Fig. 6b) and Guinier plots (Supplementary Fig. 6c) confirmed that the quality of the samples and data are suitable for further analysis. The flexibility of anti-mCD40 antibodies with different CDs was assessed by dimensionless Kratky plots[29,30]. As shown in Fig. 4a, both IgG1 and 2 dimensionless Kratky plots have maximum values close to 1.103 for $qRg = \sqrt{3}$, a feature of fully folded globular proteins. In contrast, IgG3 has a distinct dimensionless Kratky plot profile with a clearly right-shifted peak and an increased maximum value, features of proteins consisting of several domains tethered by flexible linkers[29]. These results suggest that IgG1 and 2 are relatively more rigid, whereas IgG3 is much more flexible.

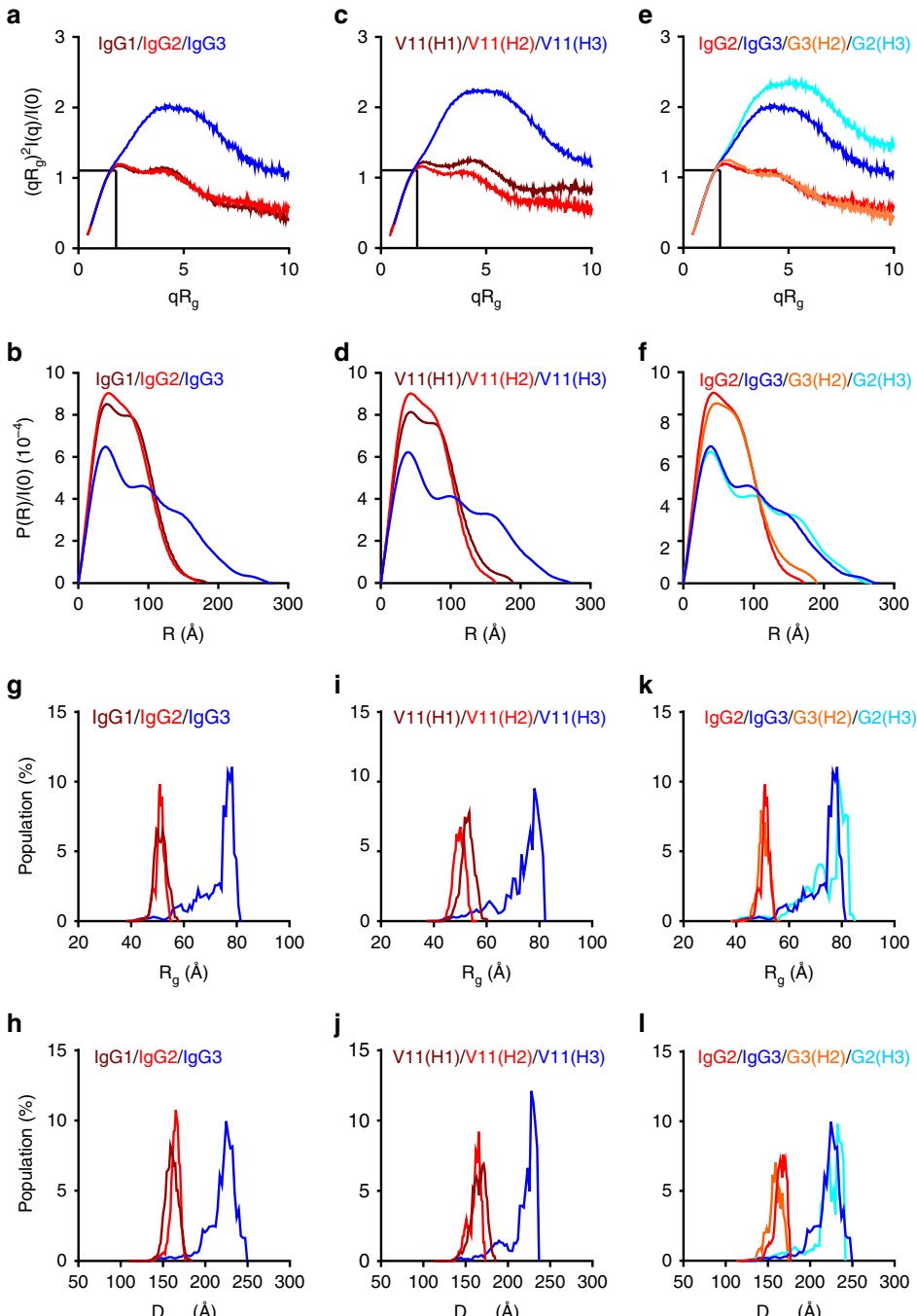

**Fig. 4** The superior flexibility of IgG3 CH1-hinge. **a–f** Dimensionless Kratky plots (**a**, **c**, **e**) and normalized interatomic distance distribution (P(R)/I(0)) (**b**, **d**, **f**) of anti-mCD40 antibodies of indicated constant domains (at the highest tested concentrations). In dimensionless Kratky plots, the position of Guinier–Kratky points ($\sqrt{3}$, 1.103) are labeled with black dashed lines. **g–l** The distribution of $R_g$ (**g**, **i**, **k**) and $D_{max}$ (**h**, **j**, **l**) in the optimized ensembles generated in EOM analysis of the indicated antibodies (at the lowest tested concentrations). A representative of two independent experiments is shown

Flexible molecules often adopt more extended conformations, which can be evaluated by the normalized distribution of interatomic distances (R) within analyzed macromolecules, i.e., $P(R)/I(0)$[29,30]. Computed distance distribution functions (P(R)) were confirmed to fit SAXS data (Supplementary Fig. 6d). As shown in Fig. 4b, IgG1 and 2 have similar interatomic distance distribution, whereas IgG3 has larger interatomic distances. IgG3 antibodies also have much larger $R_g$ and $D_{max}$ values (related to average and maximum interatomic distances, respectively) than IgG1 and 2 antibodies (Table 1). These data further support that

human IgG3 antibodies are more extended as compared to IgG1 and IgG2 antibodies.

Further analysis of a set of anti-CD40 antibodies with identical variable and Fc domains but different CH1-hinges of human IgG1–3 (V11(H1), V11(H2), and V11(H3)) showed comparable SAXS profiles and values as their corresponding natural hIgGs, including dimensionless Kratky plots (Fig. 4a, c), interatomic distance distributions (Fig. 4d, b), $R_g$ and $D_{max}$ values (Table 1). Furthermore, analysis of G2(H3) and G3(H2) chimeric antibodies showed that the different SAXS profiles and values of IgG2 and 3

**Table 1 SAXS-derived parameters**

| | G1 | G2 | G3 | V11(H1) | V11(H2) | V11(H3) | G2(H3) | G3(H2) |
|---|---|---|---|---|---|---|---|---|
| *Guinier analysis* | | | | | | | | |
| I(0) (S.D.) (A.U.) | 245.50 (0.72) | 136.69 (0.45) | 135.97 (1.62) | 161.15 (0.68) | 104.77 (0.39) | 185.75 (2.22) | 125.22 (1.78) | 130.05 (0.55) |
| $R_g$ (S.D.) (Å) | 51.54 (0.18) | 49.98 (0.21) | 69.25 (0.98) | 54.00 (0.27) | 49.06 (0.22) | 73.29 (0.92) | 73.66 (1.10) | 54.18 (0.28) |
| $q_{min}$ (Å⁻¹) | 0.0086 | 0.0086 | 0.0086 | 0.0086 | 0.0086 | 0.0086 | 0.0086 | 0.0086 |
| $qR_g$ max | 1.26 | 1.24 | 1.23 | 1.27 | 1.27 | 1.30 | 1.30 | 1.24 |
| Quality | 0.99 | 0.88 | 0.99 | 0.88 | 0.99 | 0.99 | 0.97 | 0.92 |
| *P(R) analysis* | | | | | | | | |
| I(0) (S.D.) (A.U.) | 245.7 (0.5) | 136.0 (0.3) | 143.7 (0.9) | 160.6 (0.3) | 105.4 (0.2) | 195.0 (0.9) | 129.7 (0.8) | 130.2 (0.4) |
| $R_g$ (S.D.) (Å) | 52.28 (0.13) | 50.35 (0.10) | 76.99 (0.53) | 54.84 (0.12) | 50.07 (0.10) | 80.72 (0.32) | 80.39 (0.43) | 55.93 (0.21) |
| $D_{max}$ (Å) | 184.7 | 172.2 | 278.0 | 189.5 | 164.8 | 271.9 | 273.5 | 204.5 |
| q range (Å⁻¹) | 0.0086–0.4031 | 0.0086–0.4031 | 0.0086–0.4031 | 0.0086–0.4031 | 0.0086–0.4031 | 0.0086–0.4031 | 0.0086–0.4031 | 0.0086–0.4031 |
| GNOM total estimate | 0.80 | 0.73 | 0.68 | 0.72 | 0.90 | 0.68 | 0.69 | 0.79 |
| *Molecular weight analysis (kD)* | | | | | | | | |
| $V_p$ (nm³) | 262 | 258 | 306 | 288 | 254 | 320 | 310 | 317 |
| Mr by Bayesian Inference (%) | 157 (69) | 131 (46) | 170 (29) | 138 (42) | 131 (51) | 170 (33) | 208 (37) | 170 (53) |
| Credibility interval (%) | 142–177 (99) | 116–151(94) | 127–177 (95) | 127–151 (95) | 121–151 (94) | 134–177 (95) | 99–264 (90) | 151–195 (99) |
| Mr from sequence | 145 | 144 | 155 | 145 | 145 | 155 | 155 | 144 |
| *EOM (default parameters, 10,000 models in the initial ensemble, native-like models, constant subtraction allowed)* | | | | | | | | |
| $\chi^2$ | 1.190 | 1.180 | 1.118 | 1.476 | 0.933 | 1.311 | 1.102 | 1.661 |
| No. of curves | 11 | 8 | 9 | 10 | 9 | 9 | 10 | 9 |
| $R_{flex}$ (random) (%) | -79.35 (-85.11) | -73.71 (-84.81) | -80.18 (-85.96) | -78.46 (-82.15) | -77.43 (-84.39) | -83.31 (-85.40) | -82.66 (-83.33) | -78.63 (-83.89) |
| $R_\sigma$ | 0.81 | 0.65 | 1.28 | 0.88 | 0.77 | 1.39 | 1.45 | 0.83 |

antibodies could be switched along with their CH1-hinges to a large extent (Fig. 4e, f and Table 1).

The ensemble optimization method (EOM) for SAXS data has been developed to evaluate flexibility quantitatively[31,32]. The application of this method to our SAXS data yields high-quality fits between the optimized ensemble and the experimental data of low-concentration samples (Supplementary Fig. 6e). It appears that the selected models of human IgG2 have the narrowest distribution profiles of $R_g$ (Fig. 4g) and $D_{max}$ (Fig. 4h and Supplementary Fig. 6f) among those of human IgG1–3, consistent with previous SAXS study of human IgG1 and 2 antibodies[32]. At the same time, the selected models of human IgG3 show the broadest profiles. These results suggest that human IgG2 and 3 are the least and most flexible among human IgG1–3, respectively. This notion is further supported by the ranking of $R_{flex}$ and $R_\sigma$ values, quantitative measures of flexibility: IgG3 > IgG1 > IgG2 (Table 1). Importantly, V11 variants with human IgG1–3 CH1-hinges have similar distribution profiles of $R_g$ and $D_{max}$ as human IgG1–3 antibodies, respectively (Fig. 4i–j, Supplementary Fig. 6f), as well as $R_{flex}$ and $R_\sigma$ rankings (Table 1). Furthermore, analysis of G2(H3) and G3(H2) showed that the distribution profiles of $R_g$ and $D_{max}$ (Fig. 4k, l, Supplementary Fig. 6f), $R_{flex}$ and $R_\sigma$ rankings (Table 1) of human IgG2 and IgG3 antibodies could be switched along with their CH1-hinges. Models generated by EOM analysis suggest that the long IgG3 hinge is highly flexible and can support various conformations of IgG3, V11(H3) and G2(H3) antibodies (Supplementary Fig. 6g). Further EOM analysis using high-concentration samples returned essentially the same results (Supplementary Figs. 6h–k, Supplementary Table 4). Overall, our SAXS study suggests that the flexibility of human IgG1–3 antibodies is primarily determined by their CH1-hinges and that among them human IgG2 and 3 has the least and most flexible CH1-hinges, respectively.

**The human IgG2 CH1-hinge confers the most rigidity**. To further investigate whether anti-CD40 antibodies with different hIgG CH1-hinges have varying abilities to stably hold CD40 molecules together, time-resolved fluorescence energy transfer (TR-FRET) was used to examine the relative distance between two CD40 molecules bound to these antibodies, which reflects their mobility. When anti-CD40 antibodies are mixed with CD40 labeled with Tb donor and D2 acceptor fluorochromes, referred to as "CD40-Tb" and "CD40-D2", respectively, TR-FRET signal is expected upon Tb stimulation from individual antibody molecules that simultaneously bind one CD40-Tb and one CD40-D2 on their two Ag-binding sites (Fig. 5a). According to the principle of TR-FRET, its signal is determined by the distribution of distance between CD40-Tb and CD40-D2 (referred to as "$R_{Ag}$"), and the largest distance that can trigger TR-FRET signal between Tb and D2 is two times of their Förster's radius ($2R_0$), i.e., 11.6 nm (ref. [33,34] and Cisbio.com). Because the distance between two Ag-binding sites of crystalized intact IgG1 and 4 antibodies is in the range of 12–17 nm (ref. [35,36] and Supplementary Fig. 7), no detectable TR-FRET signal is expected from anti-CD40 antibodies in these crystalized conformations (Fig. 5b). However, due to CH1-hinge flexibility of IgG antibodies in solution, the waving of Fab arms may bring bound CD40-Tb and CD40-D2 on Ag-binding sites close enough to trigger detectable TR-FRET signal (Fig. 5b). Since TR-FRET signal strength increases exponentially by the power of 6 as $R_{Ag}$ decreases from the threshold distance[34], we reasoned that antibodies with increased flexibility would have more molecules with smaller $R_{Ag}$ values and therefore increased TR-FRET signal, whereas rigid CH1-hinge would lead to reduced TR-FRET signal (Fig. 5b).

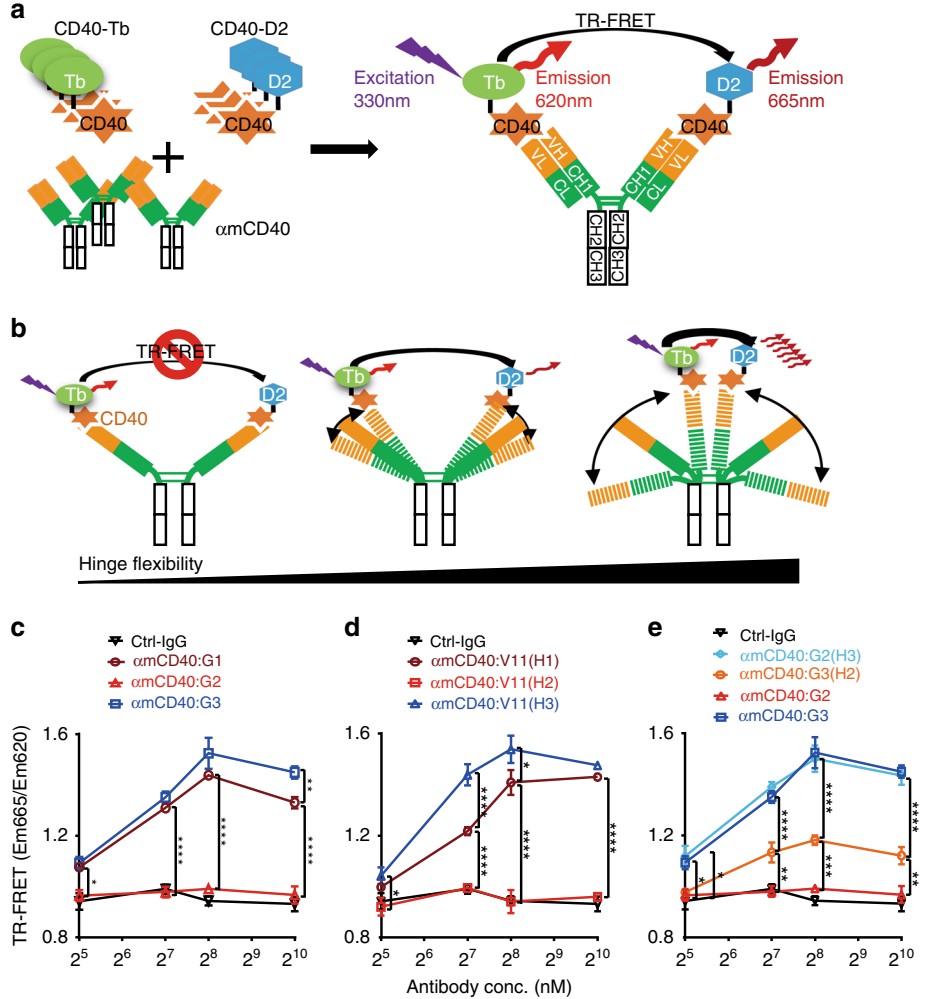

**Fig. 5** The superior rigidity of IgG2 CH1-hinge. **a** A diagram showing anti-mCD40 antibody TR-FRET. Anti-mCD40 antibody molecules mixed with CD40-Tb and CD40-D2 (mCD40 labeled with Tb donor and D2 acceptor fluorochromes, respectively) can simultaneously bind CD40-Tb and CD40-D2, and emit TR-FRET signal quantified as the relative ratio of detected 665 nm fluorescence to 620 nm fluorescence (Em665/Em620) upon stimulation. **b** A diagram of the model showing that hinge flexibility of hIgG anti-mCD40 antibodies correlates with TR-FRET signal levels. Left, anti-mCD40 antibodies with little hinge flexibility do not trigger TR-FRET signal due to the large distance between CD40-Tb and CD40-D2; middle, hinge flexibility can bring CD40-Tb and CD40-D2 close enough to trigger TR-FRET signal; right, anti-mCD40 antibodies with large hinge flexibility give rise to stronger TR-FRET signal due to more molecules with closer CD40 binding sites. **c–e** TR-FRET signal levels of anti-mCD40 antibodies of indicated constant domains. Shown are relative TR-FRET signal levels (Em665/Em620) plotted against the concentration of control IgG or the anti-mCD40 antibodies of indicated constant domains. Bars represent the mean ± SEM. *$p \leq 0.05$, **$p \leq 0.01$, ***$p \leq 0.001$, ****$p \leq 0.0001$, two-way ANOVA with Holm–Sidak's post hoc. Source data (**c–e**) are provided as a Source Data file. A representative of two independent experiments is shown

As shown in Fig. 5c, IgG3 has the strongest TR-FRET signal among human IgG1–3 anti-mCD40 antibodies, suggesting that IgG3 is the most flexible, which is consistent with our SAXS studies. Strikingly, IgG2 antibody exhibits no significant TR-FRET signal (Fig. 5c), suggesting that IgG2 is rigid enough to limit its two Fab arms from bringing bound CD40-Tb and CD40-D2 in close proximity to trigger detectable TR-FRET signal. In contrast, IgG1 antibodies have intermediate TR-FRET signals, suggesting intermediate flexibility. Further analysis of the V11 antibody set showed that V11(H1), V11(H2), and V11(H3) have similar TR-FRET signatures as IgG1–3, respectively (Fig. 5d, c), suggesting that CH1-hinge region is responsible for the different TR-FRET signalling pattern. In fact, analysis of chimeric antibodies G2(H3) and G3(H2) showed that, to a large extent, the distinct TR-FRET signature of IgG2 and 3 antibodies could be switched along with their CH1-hinges (Fig. 5e). These TR-FRET results are not only consistent with our SAXS studies, but also allow better distinction between IgG2 and other IgG subclasses.

Taken together, our SAXS and TR-FRET studies of hIgG anti-mCD40 antibodies suggest that IgG3 CH1-hinge confers the most flexibility, whereas IgG2 CH1-hinge confers the most rigidity. The inverse correlation between the agonism and flexibility of these antibodies suggests that IgG CH1-hinge flexibility is detrimental to agonism.

**Selected CH1-hinge and Fc synergize to improve hIgG agonism.** To test whether modulating the rigidity/flexibility of IgG hinge can modulate its agonistic potency, we inserted a short flexible linker ("GSGSGS") into the hinge of human IgG2 anti-mCD40 antibodies to generate the IgG2(GS)$_3$ variant. IgG2(GS)$_3$ antibodies retained mCD40 binding (Supplementary Fig. 1b) and displayed significantly stronger TR-FRET signal as compared to unmutated IgG2 control antibodies (Fig. 6a), an indication of increased hinge flexibility. Importantly, IgG2(GS)$_3$ anti-mCD40 antibodies displayed clearly reduced immunostimulatory

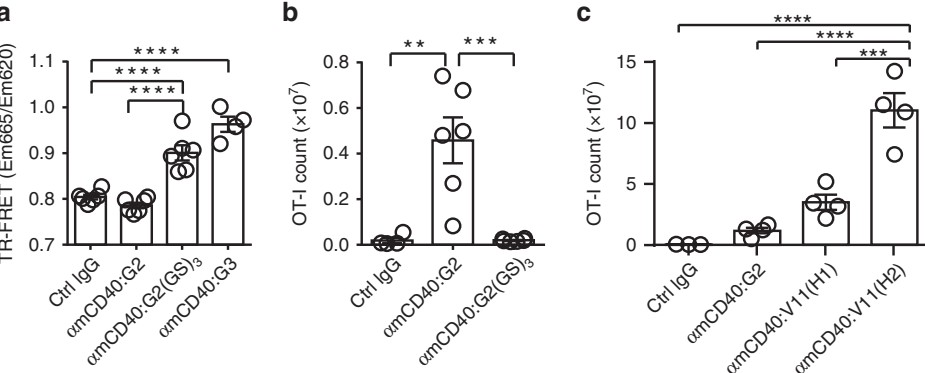

**Fig. 6** The synergy between FcγRIIB-binding Fc and rigid CH1-hinge. **a** Relative TR-FRET signal levels of anti-mCD40 antibodies of indicated constant domains (G2(GS)$_3$ is a human IgG2 constant domain variant with a "GSGSGS" insertion in the hinge (Supplementary Table 2)) quantified as in Fig. 4c–e. **b**, **c** Quantification of OT-I cells in FcγR-humanized mice treated and analyzed as in Fig. 1a, together with indicated control or anti-mCD40 antibodies of indicated constant domains (10 µg per mouse). Numbers of mice: **b** four to six mice per group; **c** three mice for ctrl IgG and four mice for each other groups. In **b**, **c**, each symbol represents an individual mouse. Bars represent the mean ± SEM. $^{**}p \leq 0.01$, $^{***}p \leq 0.001$, $^{****}p \leq 0.0001$, one-way ANOVA with Holm–Sidak's post hoc. Source data (**a–c**) are provided as a Source Data file. A representative of two independent experiments is shown

activities as compared to matched control IgG2 antibodies (Fig. 6b), supporting the notion that IgG CH1-hinge flexibility is detrimental to IgG agonism and that changing rigidity/flexibility of IgG hinge can modulate its agonistic potency.

Since hIgG CH1-hinge and Fc domains have completely different mechanisms driving IgG agonism, we investigated whether the rigid IgG2 CH1-hinge can synergize with optimized Fc's to enhance IgG agonistic potency. V11(H2) combining human IgG2 CH1-hinge and V11 Fc were compared with both V11(H1) and IgG2 parental antibodies. As shown in Fig. 6c, V11(H2) anti-mCD40 antibodies have significantly improved immunostimulatory activities as compared to both parental antibodies. Importantly, V11(H2) also showed significantly better antitumour activity as compared to both IgG2 and V11(H1) parental antibodies, with higher number of tumour-free mice in the MC38 tumour model (Fig. 3a) and strongly reduced average tumour size in both the MC38 and MO4 tumour models (Fig. 3b, c). V11(H2) anti-mCD40 antibody-treated tumour-free mice also rejected MC38 re-challenge, suggesting that these mice have developed long-term immunity to MC38 tumours (Supplementary Fig. 5b). In contrast, V11(H2) anti-mCD40 antibodies displayed no antitumour activities in FcγR-deficient mice (Supplementary Fig. 5a), confirming that Fc–FcγR interactions are required for their antitumour activities. Together, these data demonstrate that human CH1-hinge selected for rigidity and Fc domains engineered for FcγRIIB engagement can synergize to enhance the immunostimulatory and antitumour activities of anti-CD40 antibodies.

**Flexible hIgG CH1-hinge is detrimental for anti-DR5 agonism.** To investigate whether the detrimental effect of the flexible IgG3 CH1-hinge also applies to other anti-TNFR agonistic antibodies, we studied agonistic anti-DR5 antibodies. Agonistic anti-DR5 antibodies can induce apoptosis by triggering DR5 signalling that activates caspase-8 and the downstream caspase-3[14]. As shown in Fig. 7a, all natural hIgG anti-DR5 antibodies failed to induce a significant percentage of Annexin-V$^+$PI$^-$ apoptotic MC38 cells in the presence of human FcγR-expressing cells, suggesting more potent CDs might be required for agonistic anti-DR5 antibodies. Consistent with our anti-CD40 antibody study, the V11 Fc with enhanced binding to human FcγRIIB conferred clearly improved pro-apoptotic activity in anti-DR5 antibodies when treating MC38 cells co-cultured with human FcγR-expressing cells

(Fig. 7a, b). This activity was abrogated either by co-culturing with FcγR-deficient cells or by the addition of human FcγRIIB-blocking antibody 2B6 (Fig. 7b), suggesting that human FcγRIIB engagement is required. Importantly, human IgG3 CH1-hinge can efficiently inhibit the agonistic activities of V11 anti-DR5 antibodies, as shown by the significantly reduced percentage of Annexin-V$^+$PI$^-$ and active caspase-3$^+$ cells among treated MC38 cells (Fig. 7b, c). Since agonistic anti-DR5 antibodies can trigger apoptosis in cholangiocytes and cause cholestatic liver disease[14,37], we evaluated the in vivo activities of these agonistic anti-DR5 antibodies by analyzing their hepatotoxicity effects in FcγR-humanized mice. As shown in Fig. 7d, e, V11(H1) anti-DR5 antibodies induced significantly increased serum aspartate aminotransferase (AST) levels and mortality, with all mice died within 1 week. In contrast, mice treated with V11(H3) anti-DR5 antibodies were mostly protected (Fig. 7e), suggesting that the detrimental effect of the flexible IgG3 CH1-hinge on antibody agonism is not limited to anti-CD40 antibodies.

## Discussion

Taken together, our study demonstrates that, in a physiologically relevant mouse model where human FcγRs are expressed, natural hIgG CDs confer very different levels of anti-CD40 antibody agonism, including agonistically inactive IgG3 and superior IgG2. Interestingly, despite all human IgGs requiring Fc–FcγR engagement for optimal agonistic activities, these different agonistic potencies are not due to their Fc domains, but rather to their CH1-hinge regions. Although hinge regions of natural hIgG CDs have been extensively studied, evidence supporting their direct role in IgG antibody function under physiological conditions is limited[38,39]. Our study demonstrates that in the human FcγR-expressing background, hIgG CH1-hinge is directly implicated in regulating antibody agonistic function, and has a dominant contribution to the observed divergent agonistic potency of nature hIgG CDs.

We established an inverse correlation between hIgG CH1-hinge flexibility and their ability to confer agonism, with the least potent human IgG3 CH1-hinge being the most flexible and the most potent IgG2 CH1-hinge being the most rigid. We also showed that changing the rigidity/flexibility of IgG hinges can modulate its agonistic potency. How exactly the rigidity of hIgG CH1-hinge could be translated into the functional potency of agonistic antibodies is not clear. Since neither the rigid IgG2

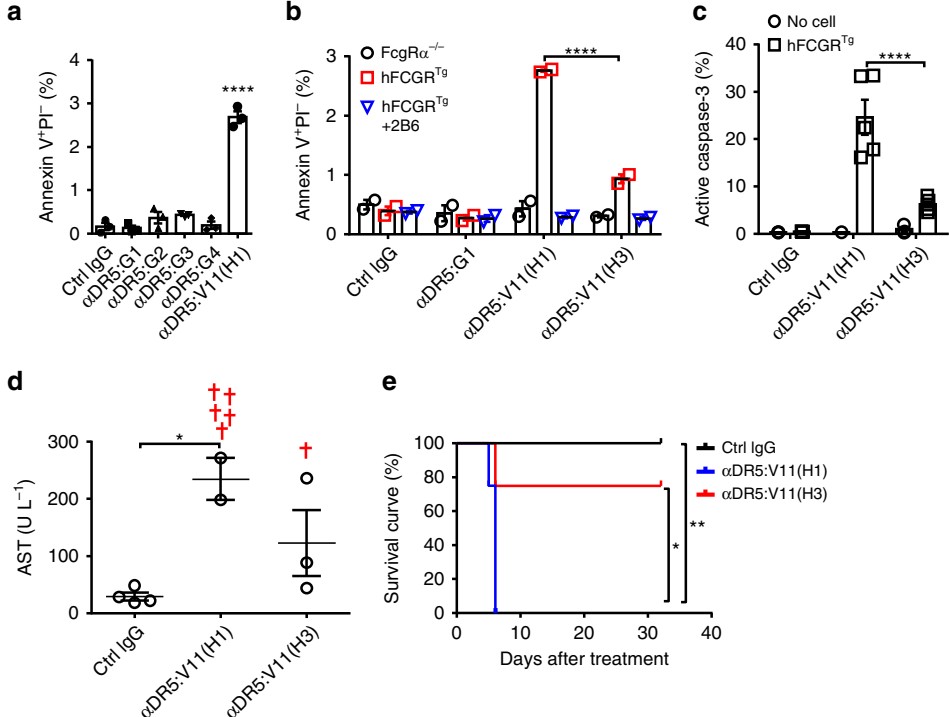

**Fig. 7** The detrimental effect of IgG3 CH1-hinge on anti-DR5 antibody agonism. **a–c** Quantification of the percentage of Annexin V+PI− (**a**, **b**) or active caspase-3-positive (**c**) apoptotic cells among MC38 cells co-cultured with FcγR-humanized (**a–c**), Fcgrα−/− (**b**), or without (**c**) mouse splenocytes for 4 h in the presence of indicated control IgG or anti-DR5 mAbs (**a–c**) with/without human FcγRIIB-blocking antibody 2B6 (**b**). **d** Quantification of AST levels in FcγR-humanized mice treated with indicated control or anti-DR5 antibodies 6 days earlier (crosses symbolize mortality). **e** Survival curve of mice in (**d**). Numbers of mice: **d**, **e** four mice per group except for seven mice in the αDR5:V11(H1) group. Bars represent the mean ± SEM. $^*p \leq 0.05$, $^{**}p \leq 0.01$, $^{***}p \leq 0.001$, $^{****}p \leq 0.0001$, one-way ANOVA Sidak's multiple comparisons test (**a**), two-way ANOVA Sidak's multiple comparisons test (**b**, **c**), and Gehan–Breslow–Wilcoxon test (**d**). Source data are provided as a Source Data file. A representative of three (**a–c**), or two (**d**, **e**) independent experiments is shown

CH1-hinge nor the potent V11 Fc can independently drive anti-CD40 or anti-DR5 agonism, we speculate that both IgG CH1-hinge and Fc contribute to antibody-mediated in vivo clustering and crosslinking of the targeted receptors, a proposed mechanism explaining how murine IgG Fc-FcγR engagement contributes to antibody agonism[11,15]. One possibility is that antibodies with rigid CH1-hinges are more efficient in clustering the targeted CD40 receptors and promoting the transition from inactive monomeric CD40 into active multimeric forms, a process that requires overcoming an energy barrier[40,41]. Antibodies with flexible hinges, on the other hand, might be inefficient in holding targeted receptors to overcome such an energy barrier to become active multimeric forms, which mimics the situation where a flexible lever is not as efficient as a rigid one to move a heavy object. Interestingly, engineered extracellular matrix binding properties of anti-CD40 antibodies have also been shown to enhance their agonistic function, likely by increasing injection-site tissue retention and promoting antibody cross-link[42].

Human IgG2 has two major conformations referred to as "H2A" and "H2B", and the "H2B" form has been reported to be more potent than the "H2A" form in driving anti-CD40 antibody agonism under the mouse FcγR background in some cases[20–22]. Interestingly, the "H2B" form has been suggested to be more compact as compared to the "H2A" form and has been proposed to confer better anti-CD40 antibody agonism because it may promote closer packing of targeted CD40 molecules[21,22]. However, our TR-FRET studies showed that human IgG2 antibodies have the weakest TR-FRET signals, indicating that their Fab arms do not have the closest distances. We speculate that the more compact "H2B" form is also more rigid as compared to the less

potent "H2A" form given that flexible molecules often adopt more extended conformations[29,30]. Our data suggest that the conformation dynamic of hIgG antibodies, instead of their specific conformation, is more relevant to their mode of action in regulating anti-CD40 antibody agonism. Furthermore, earlier studies using fluorescence anisotropy have suggested that mouse IgG1 has a more rigid hinge than mouse IgG2a[19], and we and others have shown that mouse IgG1 confers stronger agonism than mouse IgG2a in an FcγRIIB-dependent manner[10,11]. Therefore, it is possible that mouse IgG agonism is also regulated by both Fc–FcγR interactions and hinge rigidity.

Our finding of the detrimental effect of IgG hinge flexibility on anti-CD40 antibody agonism is distinct from the previously reported beneficial effects of open and flexible IgG hinges on the neutralizing activity of bi-specific anti-HIV-1 antibodies and on the capacity of IgG antibodies to fix complement[19,43]. These reported beneficial effects are likely due to increased accessibility of flexible antibodies to their specific binding epitopes on antigens. In addition, recent mapping of binding epitopes of a panel of anti-CD40 antibodies showed a correlation between agonistic activity and binding epitopes, in a way that antibodies with binding epitopes far away from cell membrane are more potent, likely due to better accessibility of antibody Fc to FcγRs[22]. In this regard, because different antibody-antigen epitope pairs may have different topological features, it is possible that extreme rigidity might be detrimental to the agonistic activities of topologically restricted antibodies. For these antibodies, there could be different optimal hinge flexibility levels that support: (1) efficient binding of antibody Fab to targeted antigen epitopes; (2) efficient binding of

antibody Fc to FcγRIIB; (3) efficient activation of targeted antigens upon binding to both antigen and FcγRIIB by antibody Fab and Fc, respectively.

It is important to note that antibody agonism can be modulated through both Fc and CH1-hinge optimization for both anti-CD40 and -DR5 antibodies, and for perhaps agonistic antibodies targeting other TNFR superfamily members as well, which may help the design of agonistic antibodies for clinical application. So far, therapeutic agonistic antibodies targeting TNFR superfamily members are still under development. But most studies use natural hIgG CDs and have stayed away from engineering the hinge and Fc domains. Our findings not only highlight the importance of hinge rigidity and selective FcγR-engagement in antibody agonistic function, but also strengthen the need for newer strategies, as compared with those currently developed for clinical trials, to modulate antibody agonism for improved clinical application.

## Methods

**Mice.** FcγR-deficient (FcγRα$^{-/-}$) and -humanized (FcγRα$^{-/-}$/hFcγRI$^+$ /hFcγRIIA$^{R131+}$/hFcγRIIB$^+$/hFcγRIIIA$^{F158+}$/hFcγRIIIB$^+$, or "hFCGR$^{Tg}$") mice[23], Fcgr2b-deficient (Fcgr2b$^{-/-}$) and -humanized (Fcgr2b$^{-/-}$hFcγRIIB$^{Tg}$) mice[10], human CD40/FcγR-transgenic mice[20] and OT-I mice[44] have been described previously and were kindly provided by Dr Jeffrey Ravetch (The Rockefeller University). For FcγR-humanized mice and human CD40/FcγR-transgenic mice, both bred mice and bone marrow chimaera mice derived from their bone marrow cells were used and confirmed to give the same results. To generate bone marrow chimeric mice, 8–10-week-old wild-type C57BL/6 mice (SLAC, Shanghai, China) were lethally irradiated at 8 Gy using RS 2000pro X-ray biological Irradiator (Rad Source Technologies, Inc., USA) and transferred with $2 \times 10^6$ bone marrow donor cells through tail vein injection. Two months after transplantation, peripheral blood of the bone marrow reconstituted mice were collected, and the reconstitution levels were confirmed to be over 95% in B cells and CD11b$^+$ cells by flow cytometry analysis of human FcγRIIA/B expression. All mice were bred and maintained in the specific-pathogen-free animal facility at the Department of Laboratory of Animal Science, Shanghai Jiao Tong University School of Medicine. All animal care and study were performed in compliance with institutional and NIH guidelines and had been approved by SJTUSM Institutional Animal Care and Use Committee (Protocol Registry Number: A-2015-014).

**Antibodies.** 1C10-derived anti-mouse CD40 antibodies, anti-human CD40 antibodies (clones 21.4.1 and 3.1.1 of Patent No.: US 7,338,660) and MD5-1-derived anti-mouse DR5 antibodies of the different heavy chain CDs were produced using published protocols[10,14]. Briefly, anti-CD40 and -DR5 antibody heavy chain expression constructs were generated either by subcloning human IgG CD sequences into a mammalian expression vector with 1C10 and DR5 heavy chain gene variable domains, respectively[10,14], or by site-directed mutagenesis. Human IgG1 CD sequences have been described previously[10]. Human IgG2–4 CD sequences were obtained by gene synthesis (Biosune, Shanghai, China) based on human IgG sequences in the IMGT database (http://www.imgt.org/). Sequences of chimeric CDs G1(H2), G3(H2), G4(H2), G2(H1), and G2(H3) were also synthesized based on IMGT sequences where "G1–G4" refer to the CH2–CH3 regions of IgG1–4 heavy chain CDs, respectively, and "H1–H3" refer to the CH1-hinge regions of IgG1–3 heavy chain CDs, respectively. V11(H1) is a previously described human IgG1 heavy chain CD variant carrying G237D/P238D/H268D/P271G/ A330R mutations[25]. V11(H2) and V11(H3) use the CH1-hinge of human IgG2 and IgG3 heavy chain CDs, respectively, together with the CH2–CH3 of V11 variant. V11(H1)–V11(H3) sequences were synthesized. IgG2(N297A) sequences were obtained by site-directed mutagenesis using specific primers (Biosune, Shanghai, China) with the QuikChange® Site-Directed Mutagenesis Kit (Stratagene) according to the manufacturer's instructions. Anti-CD40 and -DR5 antibody light chain expression constructs have been described previously[10,14]. To produce antibodies, antibody heavy and light chain expression vectors were transfected transiently into 293T cells. The secreted antibody in the supernatant was purified by protein G Sepharose 4 Fast Flow (GE Healthcare), dialyzed into phosphate-buffered saline (PBS). LPS (endotoxin) levels were analyzed by the Limulus amebocyte lysate assay (Thermo Scientific) and confirmed to be <0.1 EU μg$^{-1}$. Antibody preparations were subjected to SEC analysis to evaluate the levels of multimeric aggregates and antibodies without discernable aggregations were used. Primers used for mutagenesis are: IgG2(N297A) heavy chain (G2N297Af 5′ CGGGAGGAGCAGTTCGCCAGCACGTTCCGTGTG3′; G2N297Ar 5′ CACACGGAACGTGCTGGCGAACTGCTCCTCCCG3′); IgG2(GS)$_3$ heavy chain: (IgG2(GS)$_3$f 5′GGTAGCGGAAGCGGTAGTTGTTGTGTCGAGTGCCCACCG3′; IgG2(GS)$_3$r 5′ACTACCGCTTCCGCTACCTTTGCGCTCAACTGTCTTGTC3′).

**CD40-binding ELISA and FACS.** Binding of anti-CD40 antibodies to mouse CD40 was analyzed by ELISA. A 96-well ELISA plate (Nunc) was immobilized with 0.1 μg ml$^{-1}$ recombinant mouse CD40 protein (Novoprotein, China) at 4 °C overnight. After thoroughly aspirating the solution, wells were washed three times with 1× PBST (PBS with 0.1% Tween-20) and blocked with PBS/1%bovine serum albumin (BSA) for 2 h, followed by another three washes. Serially diluted control IgG or anti-CD40 antibodies (3.16–0.00316 μg ml$^{-1}$) were then added and incubated for 1 h. After washing for three times, horseradish peroxidase-conjugated detection antibody (anti-human IgG Fc-HRP, Bethyl Laboratories, 100 ng ml$^{-1}$) was added and incubated for 1 h. The plate was then washed and developed with TMB peroxidase substrate (KPL) in the dark for 20–40 min. The absorbance was determined at 650 nm using Multiskan™ GO Microplate Spectrophotometer (Thermo Scientific™, USA). All the procedures after coating were performed at room temperatures. For competitive FACS, splenocytes of FcγRα$^{-/-}$ mice were stained with CF640R conjugated 1C10 antibody in the presence of various amounts of control IgG or anti-CD40 antibodies, and analyzed for 1C10-CF640R staining in B220$^+$ cells using a BD LSRFortessa™ X-20 analyzer (BD Biosciences). In order to analyze the relative binding epitope of 21.4.1 and 3.1.1 anti-human CD40 antibodies, human CD40 ECD (Sino Biological, China) is immobilized on the ELISA plate, and incubated with control, 21.4.1 or 3.1.1 antibodies after blocking and washing, which is followed by washing and incubation with biotinylated 21.4.1, 3.1.1, and human CD40L (Sino Biological, China), respectively. The binding of these biotinylated proteins to immobilized CD40 ECD is detected by streptavidin–HRP and TMB reactions.

**FcγR-binding ELISA.** ELISA plates were coated with 100 μl of 2 μg ml$^{-1}$ anti-CD40 antibodies of various CDs in PBS (PH 7.4) overnight at room temperature. On the next day, the plates were blocked at room temperature for 2 h with 200 μl of 1% BSA in PBS per well and washed 2 times with PBS containing 0.05% Tween-20 (PBST). Then 100 μl of biotin-conjugated FcγRs (Sino Biological, China), 1:4 serially diluted from 1 μg ml$^{-1}$, were added. After 1 h of incubation, the plates were washed 3 times with PBST and 100 μl of PBS containing 1:5000 diluted streptavidin–HRP (BD Biosciences) was added and incubated for 1 h. After washing four times, 100 μl of TMB substrate (KPL) was added per well, and the absorbance at 650 nm was recorded.

**OVA-specific CD8$^+$ T-cell response.** Mice were adoptively transferred with CD45.1$^+$ splenic OT-I cells ($2 \times 10^6$ cells in 200 μl PBS per mouse) via tail vein injection on day-1, and immunized through intraperitoneal injection with 2 μg of DEC-OVA[45], in the presence or absence of control or anti-CD40 antibodies (3.16–100 μg per mouse, as specified in Figure legends), and/or 2B6 (150 μg per mouse). On day 6, spleen cells were harvested, and after lysing red blood cells, the single-cell suspension was stained with anti-CD4 (clone RM4-5), anti-CD8 (clone 53-6.7), anti-CD45.1 (A20), anti-TCR-Vα2 (B20.1) to quantify OVA-specific OT-I CD8$^+$ T cells. OT-I CD8$^+$ T cell is defined as CD45.1$^+$CD8$^+$TCR-Vα2$^+$ cells (Supplementary Fig. 8a). For intracellular IFN-γ staining, spleen cells were cultured for 1 h in media (RPMI with 10% fetal bovine serum (FBS), 1% Pen–Strep, 10 mM HEPES, 50 μM 2-Mercaptoethanol) with 1 μg ml$^{-1}$ anti-CD28 antibody and 1 μg ml$^{-1}$ OVA peptide SIINFEKL at 37 °C with 5% CO$_2$. Then Brefeldin A was added to a final concentration of 10 μg ml$^{-1}$, and the splenocytes were cultured for an additional 5 h. Cultured spleen cells were stained for surface CD4 (clone RM4-5) and CD8 (clone 53-6.7), then intracellular IFN-γ (anti-IFN-γ clone, XMG1.2) according to the manufacturer's protocol (BD Biosciences) and the flow cytometry is analyzed on BD FACSCanto II (BD Biosciences) (Supplementary Fig. 8b).

**Flow cytometry.** For surface staining, spleens were harvested, and single cell suspensions were prepared and depleted of erythrocytes. About $1 \sim 4 \times 10^6$ splenic cells were resuspended in 50 μl FACS buffer (1xPBS with 0.5% FBS, 2 mM EDTA, and 0.1% NaN3) with staining antibody and incubated on ice for 15 min. Then the cells were washed twice by FACS buffer and resuspended in 200 μl FACS buffer with DAPI or 7AAD and analyzed by flow cytometry. For the intracellular IFN-γ staining, an additional staining step was performed using Cytofix/Cytoperm™ Fixation/Permeabilization Solution Kit (BD Biosciences) according to the manufacturer's instructions. For reconstitution level analysis of hFCGR$^{Tg}$ chimaera mice, heparinized blood was collected from the orbit of mice and stained with anti-CD19 (clone 1D3), anti-CD11b (clone M1/70), anti-human CD32 (clone FLI8.26), and anti-human CD40(clone 5C3) to analyze the reconstitution level of human FcγR and human CD40.

**Small-angle X-ray scattering (SAXS).** Information on the condition of SAXS data collection and analysis is summarized in Supplementary Table 3. In brief, SAXS data were collected at the beamline BL19U2 equipped with Pilatus 1 M detector (DECTRIS Ltd) at National Center of Protein Science Shanghai (NCPSS) at Shanghai Synchrotron Radiation Facility (SSRF)[46]. Monomeric anti-CD40 antibodies were purified and verified by Size-Exclusion Chromatography (SEC). Serially diluted antibody samples (0.4–2.8 mg ml$^{-1}$) were collected. All samples were in HBS buffer (150 mM sodium chloride, 10 mM HEPES PH 7.4). 60 μl of each sample was continuously passed through a capillary tube exposed to a 240 × 80 μm X-ray beam. Data reduction, normalization for beamline intensity, and

buffer subtraction were carried out using the BioXTAS RAW software (Version 1.2.1) developed by Cornel High Energy Synchrotron Source (CHESS). SAXS data analysis was carried out using software from the ATSAS program suite (Version 2.8.4 (r10553))[47]. SAXS data obtained from samples with the highest concentrations were used for Guinier analysis, P(R) analysis, and Kratky plots, whereas data obtained from samples with both the highest and lowest concentrations were used for the EOM[31]. The radius of gyration ($R_g$) and $I_0$ were calculated by the "Autorg" ("Radius of Gyration") function[48] of the program PRIMUSQT from the ATSAS program suite (version 2.8.4)[49]. The pair distribution function P(R) and the maximum particle dimension ($D_{max}$) were determined using GNOM[50] (the "Distance Distribution" function of the program PRIMUSQT[49]). Porod volume ($V_P$) values were also estimated using the "Distance Distribution" function of the program PRIMUSQT[49]. Molecular mass (Mr) values were calculated using the Bayesian Inference approach[51] (the "Molecular Weight" module of the program PRIMUSQT[49]).

To apply the EOM[31], EOM 2.1, to the SAXS data of human IgG antibodies, a published method was used[32]. In brief, each antibody was considered as 5 rigid bodies connected by flexible linkers: 2 fixed CPPC fragments extracted from the hinge of PDB 1HZH[35] simulating the disulfide bond, 2 Fab and 1 Fc domains. The extracted CPPC fragments were also used as models of the second "CPRC" in the IgG3 hinge region (Supplementary Table 2). PDB files for Fab and Fc domains were either extracted from crystal structures (PDB 1HZH[35] for IgG1 Fc; PDB 4HAG[52] for IgG2 Fc) or generated by homology modelling using SWISS-MODEL Workspace[53]. For homology modelling, PDB 5W38[54] was selected as the template for IgG3 and V11 Fcs, and PDB 6AMM[55] for all IgG Fab domains. EOM was performed with the default setting.

**Time-resolved FRET (TR-FRET)**. A previously published method for TR-FRET (time-resolved FRET) was adopted with modification[56]. Mouse CD40 extracellular domain (His-tagged, Novoprotein, China) was labeled with Terbium (Tb) and D2 (HTRF® chemistry, Cisbio Bioassays, China), respectively to obtain CD40-Tb (1.5 Tb per CD40) and CD40-D2 (0.3 D2 per CD40). CD40-Tb, CD40-D2, and control or anti-CD40 antibodies were diluted in TPBS-BSA (5x PBS + 0.2%BSA + 0.05% Tween-20) to optimized concentrations and mixed to a final volume of 20 μl in ProxiPlateTM-384 F Plus 384-Well plates (PerkinElmer, part number: 6008260). The final concentration of mouse CD40-Tb was 2.6 nM, and the final concentration of mouse CD40-D2 was 41.6 nM (Conc. Ratio, CD40-Tb: CD40-D2 = 1:16). Control and anti-mouse CD40 monoclonal Abs were 2-fold serially diluted from 512 nM to 4 nM. Plates were incubated at room temperature for 1 h and then read for TR-FRET signal using a Synergy neo microplate reader (BioTek Instruments, Inc., USA) with the following setting: excitation at 330 nm followed by a delay of 50 μs before recording fluorescent counts for 400 μs with 620 nm (for Tb) and 665 nm (for D2) emission filters. TR-FRET signal was analyzed as the intensity ratio "Em 665 nm/Em 620 nm".

**Distance between antigen-binding sites on antibodies**. Published structure of full-length IgG1 and IgG4 antibodies were downloaded from PDB database (human IgG1, PDB:1HZH[35]; IgG4(PDB: 5DK3[36]). The protein structure figures were generated by the PyMOL programme (http://www.pymol.org). The distances between two Fabs are measured from the apex of heavy chain CDR3 to the other in PyMOL.

**MC38 and MO4 tumour models**. MC38 is a colon adenocarcinoma tumour cell line[14] and cells were maintained in DMEM with 10% FBS, 1% Pen–Strep, 1 mM Sodium Pyruvate, 10 mM HEPES (Invitrogen). Mice were inoculated subcutaneously with $2 \times 10^6$ MC38 cells in 200 μl PBS. Tumour growth was monitored with a calliper, and tumour volume was calculated by the formula $(L_1^2 \times L_2)/2$, where $L_1$ is the shortest diameter and $L_2$ is the longest diameter of the tumours. After tumours were established 5~7 days later, mice were randomized by tumour volume and treated with two doses of control or anti-CD40 antibodies (31.6 μg/mouse/dose) via intraperitoneal injection separated by 3 days, and monitored for tumour growth. MO4 is an OVA-expressing B16F10 melanoma cell line previously described[10], which was maintained in DMEM with 10% FBS, 1%Pen–Strep, and 0.4 mg ml$^{-1}$ Geneticin (Gibco). Mice were inoculated subcutaneously with $10^7$ MO4 cells in 200 μl PBS, and tumour growth was monitored as described for the MC38 tumour model. After tumours were established, mice were randomized by tumour volume and treated with two doses of 2 μg of DEC-OVA with control or anti-CD40 antibodies (31.6 μg/mouse/dose) via intraperitoneal injection separated by 3 days, and tumour growth was measured every 3 days after initial treatment.

**In vitro pro-apoptotic activity of anti-DR5 antibodies**. MC38 cells (~80% confluent) were split into flat 96-well tissue culture plates (Thermos, catalog no. 167008) at a density of $8 \times 10^4$ cells in 200 μl complete culture media (DMEM + 10%FBS + 1%Pen/Strep) per well, and cultured overnight. After gently aspirating culture media, $4 \times 10^5$ erythrocyte-depleted splenocytes prepared from FcgRα$^{-/-}$, or hFCGR$^{Tg}$ B6 mice and, resuspended in 100 μl complete culture media, 100 μl of culture media containing 1 μg ml$^{-1}$ of control IgG (Jackson ImmunoResearch Laboratories, catalog no. 009-000-003), αDR5:hIgG1, αDR5:hIgG2, αDR5:hIgG3, αDR5:hIgG4, αDR5:hIgG V11(H1), or αDR5:hIgG V11(H3) were added with or

without 1 μg ml$^{-1}$ of 2B6. Four hour later, cells were all harvested and stained with anti-mouse CD45.2 (BD, catalog no.560691), followed by Annexin V/PI using Annexin V FITC Apoptosis Detection Kit I (BD Biosciences, catalog no.556547) or intracellular active caspase-3 (clone C92-605; BD Biosciences) staining according to manufacturer's instructions. Samples were analyzed with a BD FACSCalibur™ or LSRFortessa™ X-20 flow cytometer. MC38 cells were gated based forward, side scatters and the lack of CD45.2-expression, and analyzed for percentage of Annexin V$^+$PI$^-$ or Actived caspase-3$^+$ apoptosis cells (Supplementary Fig. 8c, d).

**Hepatotoxicity**. To study the hepatotoxic effects of anti-DR5 antibodies, mice were treated with 100 μg of anti-DR5 antibodies i.v., and then monitored for survival over 1 month 6 days after the treatment, serum aspartate aminotransferase levels were analyzed using the MaxDiscovery Aspartate Transaminase Enzymatic Assay Kit (Bioo Scientific) according to the manufacturer's instructions.

**Surface plasmon resonance (SPR)**. SPR experiments were performed with a Biacore T100 SPR system (Biacore, GE Healthcare) using a published protocol[20]. In brief, experiments were performed at 25 °C in HBS EP + buffer [10 mM HEPES (pH 7.4), 150 mM NaCl, 3.4 mM EDTA, 0.005% surfactant P20]. His-tagged soluble human FcγRIIB extracellular domains (Sino Biological Inc.) were immobilized on CM5 chips by amine coupling. Twofold serially diluted (8000–16 nM) human IgG1 and IgG2 antibodies (clone 21.4.1) were injected through flow cells for 150 s at a flow rate of 30 μL min$^{-1}$ for association followed by a 6-min dissociation phase. After each assay cycle, the sensor surface was regenerated with a 30-s injection of NaOH of optimized concentration at a flow rate of 50 μL min$^{-1}$. Background binding to blank immobilized flow cells was subtracted, and affinity constant $K_D$ values were calculated using the 1:1 binding kinetics model built in the BIAcore T100 Evaluation Software (version 1.1).

**Statistical analysis**. Statistical analyses were performed with GraphPad Prism (version 6.01, for windows) and $p$ values of less than 0.05 were considered to be statistically significant. Asterisks indicate statistical comparison with the control group unless indicated otherwise on the figures ($^*p \leq 0.05$. $^{**}p \leq 0.01$, $^{***}p \leq 0.001$, $^{****}p \leq 0.0001$).

**Reporting summary**. Further information on research design is available in the Nature Research Reporting Summary linked to this article.

## Data availability
The source data underlying Figs. 1c–g, 2a–c, e, f, 3a–c, 6a–c, 7a–e and Supplementary Figs. 1a, b, 2a–c, 3a, c, d, 4a, b, and 5a, b are provided as a Source Data file. All data generated or analyzed during this study are either included in this paper (Figures and Supplementary Information) or available upon reasonable request.

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

## Acknowledgements

We thank Dr. Jeffrey Ravetch of The Rockefeller University for providing FcγR/CD40-mutant and transgenic mice, MC38 and MO4 cells. We thank Drs. Na Li and Guangfeng Liu of NCPSS beamline BL19U2 at National Center for Protein Sciences Shanghai and Shanghai Synchrotron Radiation Facility, Shanghai, People's Republic of China, and Dr. Jiansheng Jiang at National Institute of Allergy and Infectious Diseases/National Institutes of Health, Bethesda, MD, USA for their assistance during SAXS data collection and analysis. We thank Dr. Annette Langkilde of University of Copenhagen for help with EOM. We are grateful to Drs. Fredrik Wermeling and Andy Tsun for critically reading the manuscript, and to Drs. Yuan Zhuang and Bing Su for helpful discussions. We acknowledge the assistance of staff in the Department of Laboratory Animal Science, Shanghai Jiao Tong University School of Medicine and Shanghai Institute of Immunology. This work was supported by NNSFC (projects No. 31422020, 31370934 and 31700806); X.L. is supported by China Postdoctoral Science Foundation (Grant No. 2016M591688); Y.Z. is supported by Shanghai Sailing Program (project No. 16YF1409700); H.Z. is supported by Natural Science Foundation of Shanghai (project No. 15ZR1436400) and by the Program for Professor of Special Appointment (Eastern Scholar) at Shanghai Institutions of Higher Learning (2015); this project is also supported in part by Shanghai Municipal Science and Technology Commission (project No. 19431902900). F.L. is supported by the "Shu Guang" project from Shanghai Municipal Education Commission and Shanghai Education Development Foundation and the programfor Professor of Special Appointment (Eastern Scholar) at Shanghai Institutions of Higher Learning. F.L., Y.Z., H.Z. are supported by Innovative research team of high-level local universities in Shanghai.

## Author contributions

F.L. and X.L. designed the experiments; X.L., Y.J.Z., H.S., Y.Z., X.Y., M.L. and H.Z. performed the experiments. F.L., X.L., Y.J.Z., H.S., B.L., Y.H. and T.J. analyzed the results, F.L., X.L. and Y.J.Z. wrote the paper.

## Additional information

**Competing interests:** A patent application titled "Sequences of antibody heavy chain constant domains for enhanced agonistic activities and screening method", has been filed by Shanghai Jiao Tong University School of Medicine (Chinese patent application no. 201710429281.6, pending), and the listed inventors are F.L., X.L., Y.Z., Y.J.Z., H.S. and H.Z. The remaining authors declare no competing interests.

