## [Peer Review File · Nature Communications]

Reviewer #2 (Remarks to the Author):

The development of agonistic antibodies directed against members of the TNF receptor family, e.g. costimulatory and death receptors, have faced obstacles in clinical translation, mainly due to the fact that most of these receptors require receptor clustering for activation. The authors now provide a in-depth and comprehensive study analyzing the effects of antibody isotype and hinge flexibility on the agonistic activity of an anti-CD40 antibody. The authors applied a plethora of biochemical in vitro methods combined with in vivo data from transgenic animals. In summary, they could establish the importance of hinge flexibility/rigidity and the requirement FcγR binding to induce an agonistic activity. In addition to anti-CD40 antibodies, they also included an antibody directed against DR5, confirming their results. The manuscript is well written and certainly of great interest to a broader readership in this field of research and developments.

The authors discuss that the epitope of the antibody might influence outcome of agonistic activity. I am wondering if information about the epitopes of the applied anti-CD40 antibodies is available or can be obtained and included in the manuscript.

Although the authors provide information that all antibodies were SEC purified for the SAXS study, it is essential that this purification step was also applied to the antibodies used in the other studies. If not, residual multimeric aggregates and complexes might contribute to an agonistic activity. This issue should be clarified.

The authors should provide for a better understanding a table/figure showing all the applied hinge sequences.

Please check for typos: e.g. page 14: immunostimualtory, activitieis

Reviewer #3 (Remarks to the Author):

This manuscript reports a study of the impact of Fc domains and hinge regions on the agonistic functions of human immunoglobulin G. FcγR-humanized mouse model is used to evaluate the agonistic function of multiple anti-CD40 and -DR5 antibody sets. The ability to confer agonism is correlated to the degree of flexibility of the CH1-hinge as studied using biophysical techniques, SAXS and FRET. My main point of criticism concerns the SAXS analysis which exhibit significant weaknesses and must be redone.

1. Basic data analysis

Minor remarks:

- The smallest q-value is somewhat too high. It is recalled that q_{min} must be $\ll n/D_{max}$.
- The authors do not specify if data collected at different concentrations are strictly identical and if they use in their analysis the curve extrapolated at $c=0$ or the curve measured at the highest concentration.

Main remarks concern the results in the table presented in Fig. 4G:

- How is $R_g(\text{Å})$ STDEV calculated? A value as high as 8% makes the results suspect.
 - How is M_r calculated from the Porod volume? To obtain a more reliable determination of M_r the authors have to use the module "Molecular Weight" available in PRIMUSQT which combines four concentration independent MM estimators.
 - The values of M_r extracted from the SAXS data are higher than the values calculated from sequence in almost all cases. The difference could be ascribed to the presence of larger objects in very small proportion. The contribution to the SAXS curve of these larger objects could be also responsible for the surprisingly high values of D_{max} in the case of G3, V11(H3), G2(H3) (see comment below).
- In brief, the authors should present in a revised manuscript a more complete Table following

recommendations recently reported in Trehwella et al. (2017 publication guidelines for structural modelling of small-angle scattering data from biomolecules in solution: an update, J. Trehwella et al., *Acta Cryst.* (2017). D73, 710).

2. SAXS data interpretation

The authors have to be careful while using the word "flexibility", as it is nicely explained in "Comparisons of the ability of human IgG3 Hinge Mutants, IgM, IgE, and IgA2, to form small Immune complexes: A Role for Flexibility and Geometry" (K. H. Roux and al., *J. Immunol.* 161(1998)4083). The only definitive conclusion that can be extracted from the SAXS data without modelling is that G3, V11(H3) and G2(H3) are much more extended than the other antibodies. Concluding on flexibility is much more complicated.

It is unfortunate that the authors seem unaware of recent SAXS studies of antibodies IgG (Tian et al., *J. Pharm. Sci.* 103(2014)1701, Rayner et al., *JBC* 290(2015)8420) and especially the excellent work "In-depth analysis of subclass-specific conformational preferences of IgG antibodies", X. Tian, B. Vestergaard, M. Thorolfsson, Z. Yang, H. B. Rasmussen and A. E. Langkilde, *IUCrJ* 2(2015)9. Due to the flexible linkers connecting the Fab and Fc domains, IgG antibodies adopt in solution several conformations. As explained in the above article, it is thus absolutely necessary to describe the SAXS data in terms of structural ensembles for example by using the program EOM which "quantifies" the flexibility. The main result is that IgG1 and IgG2 antibodies are both flexible in solution but adopt different types of conformations: "IgG2 adapt to fewer overall conformations in solution, while IgG1 reveals a continuum of conformations around the preferred intermediate Y-shape" as nicely represented by the fig.3 of this article. It should be noted that these results are not in agreement with the FRET data shown in the manuscript of Liu et al..

In conclusion, authors must perform new analysis of their data (or preferably of improved data devoid of any contributions due to larger objects) by using such an approach.

I look forward to seeing the results for G3, V11(H3) and G2(H3) because the very high value obtained for D_{max} leads to the suspicion that IgG3 adopts always very extended conformations, and thus are not very flexible. This would disagree with the FRET data! It is even very strange that the D_{max} value (of the order of 250-270 Å) is higher than the most extended conformations described by Tian et al. (of the order of 200 Å). I therefore believe that a small amount of larger objects could be present in the samples.

Reviewer #4 (Remarks to the Author):

This manuscript addresses an unresolved issue relating to the structure/function mechanisms that contribute to immune stimulation mediated by agonist human IgG antibodies. The authors have previously shown that human IgG1 and IgG2 agonistic anti-CD40 antibodies depend on FcγR binding, with FcγRIIB binding demonstrating improved agonism. However, others have shown that human IgG2 agonism does not depend on Fc-FcγR interactions, but rather on hinge conformation. This becomes more complicated as there is evidence that the impact of IgG2 hinge conformation may not apply to all anti-CD40 antibody clones and may not apply in FcγR-humanized mice. Therefore, the authors examine if IgG hinge and FcR-interactions impact anti-CD40 and anti-DR5 antibody agonism using FcγR-humanized mice. IgG hinge biophysical flexibility was investigated using different natural and engineered human IgG Fc's and CH1-hinges to determine their influence on agonistic function.

The authors found that different IgGs induce different agonistic responses and that IgG hinge and Fc domains together regulate agonism through specific mechanisms. They show that IgG2 antibodies are highly agonistic whilst IgG3 antibodies are inactive. This activity was primarily mediated by the biophysical flexibility of CH1-hinges, although Fc-FcγR binding was still required for optimal agonistic outcomes. The authors compare the efficacy of their anti-CD40 antibodies to unmodified antibody in murine models of colon cancer and melanoma and demonstrate that their modified anti-mouse CD40V11(H2) provides significantly improved anti-tumour efficacy (though the figures were difficult to

read). They also show that the effects of hinge rigidity/flexibility on agonism are not limited to anti-CD40 antibodies as they could be reproduced in agonistic anti-DR5 antibodies.

Overall, this is a solid and very interesting paper, worthy of publication. The authors provide a resolution for the apparently conflicting data published by themselves and others. The novelty of their findings lies in demonstrating the importance of both hinge rigidity and selective FcγR (i.e. FcγRIIB) binding in antibody agonistic function. These findings may indeed represent improved strategies to improve antibody agonism for clinical application.

Major concerns

- Animal ethics approval numbers should be provided.
- In the figure legends please state the number of mice used in each experimental group for all figures as it is sometimes difficult to see.

Re Figure 1: Please state that the N297A mutation abrogates Fc-FcγR binding on the figure or in the figure legend.

Re Supp Fig 1:

- a. When were the samples collected?
- b. There also seem to be missing data sets, e.g. from control untreated mice, and mice treated with PIGF-2123-144-aCD40.

Re: Figure 2: Please clarify what anti-hCD4021.4.1 in the legend.

Re Figure 3:

- There is a typo in the legend re (A-C) and (A-B)
- It is not clear what day 0 is. Is it the day of tumour cell inoculation, or when palpable tumours could be detected?
- How big were the tumours when treatment commenced - you cannot tell from the graphs. My main concern is that differing tumour sizes could respond differently to anti-CD40 antibody and it is difficult to tell on the graph if size impact response to the different antibodies. For example, the control IgG treated tumour sizes look like they might be slightly larger than the ones given anti-CD40:V11(H2).
- Figures 3B and C make it very difficult to see all treatment groups. Please make the figures bigger. It would also be helpful state what N2974 is to remind the reader.

Re Figures 4, 5, S6 and S7: These are very interesting data, however the methods used are out of my field of expertise.

- The legend for figure 5 needs correcting – e.g. A diagram showing anti-mCD40 antibody TR-FRET and B) A diagram of the model showing that hinge flexibility of human IgG anti-mCD40 antibodies inversely correlates with TR-FRET signal levels
- For Figure 6 it would be helpful to say what G2(Gs)3 is.

Re Figure 7:

- What do the crosses on Fig 7D symbolize?
- How can the difference between control IgG and the others be statistically significant, there are only 2 mice in one group?

Re figure S3: I found the figure confusing, I thought G2 bound FcγR2B?

Re typos and English:

1. Line 30 – should read ‘...human IgGs requiring Fc-FcγR binding...’
2. Line 158 should say ‘investigated’
3. Line 186: ‘antibodies having comparable mCD40 and human FcγR binding profiles with V11(H1) antibodies (Figs. S1B and S3B). These results demonstrate that human IgG3 CH1-hinge deprives V11

Fc of its strong agonistic potency.'

4. Line 212 Title: 'The impact of human IgG CH1-hinge and Fc domains on anti-CD40 antibody immunostimulatory activities can be translated into antitumor activities'

5. Line 330: 'Importantly, IgG2(GS)3 anti-mCD40 antibodies displayed clearly reduced immunostimulatory activities as compared to matched control IgG2 antibodies (Fig. 6B)'

6. Line 361: 'This activity was abrogated either by co-culturing with FcγR-deficient cells or by the addition of human FcγRIIB blocking antibody 2B6 (Fig. 7B), suggesting that human FcγRIIB engagement is required'

7. Line 371: 'In contrast, mice treated with V11(H3) anti-DR5 antibodies were mostly protected.'

8. Line 378: 'Interestingly, despite all human IgGs requiring Fc-FcγR engagement....'

9. Line 382: 'Our study demonstrates that in the human FcγR-expressing background, the human IgG CH1-hinge is directly implicated in regulating antibody agonistic function, and has a dominant contribution to the observed divergent agonistic potency of natural human IgG constant domains.'

10. Line 392: 'we speculate that both IgG CH1-hinge and Fc contribute to antibody-mediated in vivo clustering and crosslinking of the targeted receptors, a proposed mechanism explaining how...'

11. Line 532: 'spleens were harvested'

12. Line 587: 'MO4 is an OVA-expressing B16F10 please state this melanoma cell line previously described'

Please note that the authors' point-by-point response is italicized.

Reviewer #2 (Remarks to the Author):

The development of agonistic antibodies directed against members of the TNF receptor family, e.g. costimulatory and death receptors, have faced obstacles in clinical translation, mainly due to the fact that most of these receptors require receptor clustering for activation. The authors now provide a in-depth and comprehensive study analyzing the effects of antibody isotype and hinge flexibility on the agonistic activity of an anti-CD40 antibody. The authors applied a plethora of biochemical in vitro methods combined with in vivo data from transgenic animals. In summary, they could establish the importance of hinge flexibility/rigidity and the requirement FcγR binding to induce an agonistic activity. In addition to anti-CD40 antibodies, they also included an antibody directed against DR5, confirming their results. The manuscript is well written and certainly of great interest to a broader readership in this field of research and developments.

We thank the reviewer for the positive comments and constructive suggestions.

The authors discuss that the epitope of the antibody might influence outcome of agonistic activity. I am wondering if information about the epitopes of the applied anti-CD40 antibodies is available or can be obtained and included in the manuscript.

Yes, regarding the epitopes of the analyzed antibodies, the following information is available from previous studies (References No. 26 and 27 in the revised manuscript: Cancer Immunol

Res. 2014 Jan;2(1):19-26¹; Patent No.:US 7,338,660 (2008)²): 1) The binding site of 1C10 (the anti-mouse CD40 antibody clone used in this study) on mouse CD40 overlaps with that of mouse CD40L¹; 2) 21.4.1 (one anti-human CD40 antibody clone used in this study) does not complete with human CD40L for binding to human CD40²; 3) 3.1.1 (another anti-human CD40 antibody clone used in this study) does complete with human CD40L for binding to human CD40². We confirmed that 21.4.1 and 3.1.1 do not block each other for human CD40 binding since they have different binding epitopes on CD40 (Supplementary Fig. 3d, reproduced below as Figure R1). We also confirmed that 21.4.1 and 3.1.1 have different efficiency in blocking the binding of human CD40L to human CD40 as reported (Supplementary Fig. 3d, reproduced below as Figure R1). To clarify the point raised by the reviewer, we have included the following revision in the revised manuscript:

- 1) A new supplemental table (Supplementary Table 1, reproduced below as Table R1) summarizing the epitope information;*
- 2) New supplementary data (Supplementary Fig. 3d, reproduced below as Figure R1) confirming that 21.4.1 and 3.1.1 have different binding epitopes are added into the revised manuscript and described at lines 200~203.*
- 3) To emphasize that the impact of CH1-hinge on agonism does not depend on specific binding epitopes, a statement at lines 212~217 is revised to “These data, together with our anti-mCD40 antibody data, demonstrate that while natural human IgG constant domains require Fc-FcγR interactions to drive optimal anti-CD40 antibody agonism, the intrinsically different agonistic potency of these natural IgG constant domains, including the inactive IgG3 and the superior IgG2 constant domains, is generally due to their different CH1-hinge, not Fc or association with specific binding epitopes.”*

Figure R1 (New Supplementary Fig. 3d in the revised manuscript). Clone 21.4.1 and 3.1.1 have different binding epitopes on human CD40. Presented are binding of 21.4.1, 3.1.1 and human CD40L (hCD40L) to immobilized human CD40 extracellular domain (ECD) preincubated with control (Ctrl IgG), 21.4.1 or 3.1.1 antibodies, quantified by ELISA signals (O.D.). In brief, human CD40 ECD is immobilized on ELISA plate, and incubated with control, 21.4.1 or 3.1.1 antibodies after blocking and washing, which is followed by washing and incubation with biotinylated 21.4.1, 3.1.1 and human CD40L, respectively. The binding of these biotinylated proteins to immobilized CD40 ECD is detected by Streptavidin-HRP and TMB reactions.

Table R1 (New Supplementary Table 1 in the revised manuscript). Binding epitopes of CD40 antibodies, in relationship to CD40L binding sites.

Specificity	Clone	Binding epitope (related to CD40L binding)	References
Mouse CD40	1C10	Overlap	26
Human CD40	21.4.1	No overlap	26, 27
Human CD40	3.1.1	Overlap	26, 27

Although the authors provide information that all antibodies were SEC purified for the SAXS study, it is essential that this purification step was also applied to the antibodies used in the

other studies. If not, residual multimeric aggregates and complexes might contribute to an agonistic activity. This issue should be clarified.

We agree with the reviewer that the issue of aggregates-induced agonism needs to be clarified in the context of our findings. Although not all antibody preparations used in our study were subjected to SEC purification, we do routinely evaluate the levels of aggregates in our antibody preparations by SEC and non-reduced SDS-PAGE to assess their potential contributions to our findings. We concluded that our findings are not due to residual multimeric aggregates for the following two reasons: 1) our experiments were performed using antibody preparations without discernable aggregates as assessed by SEC (see below in Figure R2 for the SEC profile of human IgG1, 2 and 3 antibodies that were used for in vivo studies without SEC purification); 2) Antibodies with distinct agonistic activities in our study were prepared in the same way, and we did not observe any differences in their purity for monomeric antibodies. To clarify this point, a statement “Antibody preparations were subjected to SEC analysis to evaluate the levels of

multimeric aggregates and antibodies without discernable aggregations were used.” in the methods section of the revised manuscript at line 510.

**Figure R2 (Data not shown in the revised manuscript).
Size-Exclusion Chromatography (SEC) profiles of
anti-mouse CD40 antibodies (without SEC
purification) of the indicated constant domains separated
with a Superdex200 column.**

The authors should provide for a better understanding a table/figure showing all the applied hinge sequences.

As suggested by the reviewer, a new table summarizing all applied hinge sequences is now included in the revised manuscript as Supplementary Table 2 (reproduced below as Table R2).

Table R2 (Supplementary Table 2 in the revised manuscript). Hinge and Fc sequences of various human IgG constant domains.

Constant domains	C-terminal of CH1	Upper hinge	Middle hinge	Hinge length	Fc
G1	NTKVDKRV	EPKSCDKTHT	CPPCP	15	G1
G2	NTKVDKTV	ERK	CCVECPPCP	12	G2
G2(GS) ₃	NTKVDKTV	ERKGSGSGS	CCVECPPCP	18	G2
G3(H2)	NTKVDKTV	ERK	CCVECPPCP	12	G3
G3	NTKVDKRV	ELKTPLGDTTHT	CPRCP(EPKSCDTPPPCPRCP) ₃	62	G3
G2(H3)	NTKVDKRV	ELKTPLGDTTHT	CPRCP(EPKSCDTPPPCPRCP) ₃	62	G2
G4	NTKVDKRV	ESKYGPP	CPSCP	12	G4
V11(H1)	NTKVDKRV	EPKSCDKTHT	CPPCP	15	V11
V11(H2)	NTKVDKTV	ERK	CCVECPPCP	12	V11
V11(H3)	NTKVDKRV	ELKTPLGDTTHT	CPRCP(EPKSCDTPPPCPRCP) ₃	62	V11

Please check for typos: e.g. page 14: immunostimulatory, activities

Thank the reviewer for pointing out the typos, which are all revised accordingly.

Reviewer #3 (Remarks to the Author):

This manuscript reports a study of the impact of Fc domains and hinge regions on the agonistic functions of human immunoglobulin G. FcγR-humanized mouse model is used to evaluate the

agonistic function of multiple anti-CD40 and -DR5 antibody sets. The ability to confer agonism is correlated to the degree of flexibility of the CH1-hinge as studied using biophysical techniques, SAXS and FRET.

My main point of criticism concerns the SAXS analysis which exhibit significant weaknesses and must be redone.

We thank the reviewer for the in-depth review and very helpful comments on our SAXS data. As suggested by the reviewer, we have redone all SAXS data analysis and presented updated and new results in the revised manuscript, including the updated results for Guinier analysis, $P(R)$ analysis, Kratky plots, and new results for EOM analysis. These results are presented in the updated and/or new Supplementary Table 3, Table 1, Fig. 4, and Supplementary Fig. 6, which are reproduced below as Table R3, Table R4, Figure R3, and Figure R4, respectively. The reanalysis strengthened our original conclusions (see below and the revised manuscript).

Table R3 (New Supplementary Table 3 in the revised manuscript). The condition of SAXS Data-collection and analysis

(a) SAXS data-collection parameters	
Instrument/data processing	The beam line BL19U2 of National Center for Protein Science Shanghai (NCPSS) at Shanghai Synchrotron Radiation Facility (SSRF)
Wavelength (Å)	1.033
Beam size (mm)	0.40 x 0.15
q measurement range (Å ⁻¹)	0.0086-0.40307
Normalization	To transmitted intensity by beam-stop counter
Monitoring for radiation damage	To reduce the radiation damage, a flow cell made of a cylindrical quartz capillary with a diameter of 1.5 mm and a wall of 10 μm was used.
Exposure time	Continuous 1 second

Sample configuration	Sixty μ l of each sample was continuously passed through a capillary tube exposed. Measurements were carried out at 3 different concentrations in all cases (0.4-2.8mg/ml)
Sample temperature ($^{\circ}$ C)	25

(b) Software employed for SAXS data reduction, analysis, interpretation, and modelling

SAXS data reduction	BioXTAS RAW (version 1.2.1)
Basic analyses: R_g , I_0 , Guinier	The AUTORG program (the "Radius of Gyration" function of the program PRIMUSQT from the ATSAS program suite (Version 2.8.4))
Basic analyses: $P(r)$, V_P	The GNOM program (the "Distance Distribution" function of the program PRIMUSQT from the ATSAS program suite (Version 2.8.4))
Ensemble modelling	The Ensemble Optimization Method (EOM 2.1)
Homology modelling	The SWISS-MODEL Workspace
PDB file processing	The PyMOL Molecular Graphic System (version 1.7.2.1, Schrodinger LLC.)

Table R4 (Table 1 in the revised manuscript (Updated based on previous Fig. 4G)) SAXS-derived parameters.

	G1	G2	G3	V11(H1)	V11(H2)	V11(H3)	G2(H3)	G3(H2)
Guinier analysis								
I(0) (S.D.) (A.U.)	245.50 (0.72)	136.69 (0.45)	135.97 (1.62)	161.15 (0.68)	104.77 (0.39)	187.44 (2.22)	127.85 (2.33)	130.05 (0.55)
R _g (S.D.) (Å)	51.54 (0.18)	49.98 (0.21)	69.25 (0.98)	54.00 (0.27)	49.06 (0.22)	74.06 (0.88)	75.54 (1.52)	54.18 (0.28)
q _{min} (Å ⁻¹)	0.0086	0.0086	0.0086	0.0086	0.0086	0.0086	0.0086	0.0086
qR _g max	1.26	1.24	1.23	1.27	1.27	1.28	1.26	1.24
Quality	0.99	0.88	0.99	0.88	0.99	1.00	1.00	0.92
P(R) analysis								
I(0) (S.D.) (A.U.)	245.7 (0.5)	136.0 (0.3)	143.7 (0.9)	160.6 (0.3)	105.4 (0.2)	195.0 (0.9)	129.7 (0.8)	130.2 (0.4)
R _g (S.D.) (Å)	52.28 (0.13)	50.35 (0.10)	76.99 (0.53)	54.84 (0.12)	50.07 (0.10)	80.72 (0.32)	80.39 (0.43)	55.93 (0.21)
D _{max} (Å)	184.7	172.2	278.0	189.5	164.8	271.9	273.5	204.5
q range(Å ⁻¹)	0.0086- 0.40307	0.0086- 0.40307	0.0086- 0.40307	0.0086- 0.40307	0.0086- 0.40307	0.0086- 0.40307	0.0086- 0.40307	0.0086- 0.40307
χ ² (GNOM total estimate)	80.0	73.4	68.8	71.9	90.0	68.2	68.5	79.0
V _P (nm ³)	262	258	306	288	254	320	310	317
Mr from V _P (KD)	152	149	177	167	147	185	179	184
Mr from sequence (KD)	145	144	155	145	145	155	155	144
EOM (default parameters, 10,000 models in the initial ensemble, native-like models, constant subtraction allowed)								
χ ²	1.190	1.181	1.118	1.476	0.935	1.311	1.102	1.661
No. of curves.	11	12	9	10	7	9	10	9
R _{flex} (random) (%)	~ 79.35 (~ 85.11)	~ 77.07 (~ 85.20)	~ 80.18 (~ 85.96)	~ 78.46 (~ 82.15)	~ 78.62 (~ 83.43)	~ 83.31 (~ 85.40)	~ 82.66 (~ 83.33)	~ 78.63 (~ 83.89)
R _δ	0.81	0.79	1.28	0.88	0.83	1.39	1.45	0.83

Fig. 4

Figure R3 (The updated Figs. 4a-f and new Figs. 4g-l in the revised manuscript). The superior flexibility of IgG3 CH1-hinge. (a-f) Dimensionless Kratky plots (a, c, e) and normalized interatomic distance distribution ($P(R)/I(0)$) (b, d, f) of anti-mCD40 antibodies of indicated constant domains. In dimensionless Kratky plots, the position of Guinier–Kratky points ($\sqrt{3}$, 1.103) are labelled with black dashed lines. (g-l) The

R_g (g, i, k) and of the D_{max} (h, j, l) distributions in the optimized ensembles generated in EOM analysis of the indicated antibodies.

Supplementary Fig. 6

Supplementary Fig. 6 (continued)

Supplementary Fig. 6 (Continued)

Supplementary Fig. 6 (continued)

Figure R4 (The Supplementary Fig 6 in the revised manuscript, including the updated Supplementary Figs. 6c-d and new Supplementary Figs. 6e-g). Small angle X-ray scattering (SAXS) analysis of anti-CD40 antibodies. (a) Size-Exclusion Chromatography (SEC) profiles of anti-mouse CD40 antibodies of the indicated constant domains separated with a Superdex200

column at a flow rate of 0.5 ml/min. **(b-c)** Scattering intensity plots (Log $I(q)$ versus q) **(b)** and Guinier plots ($\ln I(q)$ vs q^2) **(c)** of serial-diluted anti-CD40 antibodies of indicated constant domains were collected at labelled concentrations. **(d)** Fitting of computed (solid lines) to experimental (dots) scattering intensity (Log $I(q)$) plots of serially diluted antibody samples in (a) and (b) (colour-coded as in (a) and (b): blue, high concentration; red, middle concentration; black, low concentration) in distant distribution analysis ($P(R)$) by GNOM. Quality scores are labelled. **(e)** Fits between the calculated scattering curve from the best ensemble (red line, selected by EOM) and the experimental data (black line). **(f)** D_{max} distributions of the ensemble pool (black) and the optimized ensembles (red) of indicated antibodies. **(g)** Models composing the best fitting ensemble generated by the EOM analysis of indicated antibodies, with the Fc domains pointing to the lower left direction and the overall occurrence noted.

1. Basic data analysis

Minor remarks:

- The smallest q -value is somewhat too high. It is recalled that q_{min} must be $\ll \pi/D_{max}$.

The q_{min} in our SAXS data is 0.0086 \AA^{-1} , which is smaller than π/D_{max} of all samples ($0.0114 \sim 0.0168 \text{ \AA}^{-1}$). The q_{min} values in our SAXS data (Supplementary Fig. 6b in the revised manuscript) are also consistent with the published data (X. Tian, B. Vestergaard, M. Thorolfsson, Z. Yang, H. B. Rasmussen and A. E. Langkilde, *IUCrJ* 2(2015)9, Figure 1b)³ cited by the reviewer, which has a q_{min} of 0.009 \AA^{-1} .

[REDACTED]

- The authors do not specify if data collected at different concentrations are strictly identical and if they use in their analysis the curve extrapolated at $c=0$ or the curve measured at the highest concentration.

Data collected at different concentrations give consistent results in Guinier analysis (Supplementary Fig. 6c and Table 1 in the revised manuscript (reproduced above in Figure R4c and Table R4)), and all other analyses performed, including $P(R)$ analysis, Kratky plots and EOM analysis (Fig. 4, Table 1, and Supplementary Fig. 6 in the revised manuscript (reproduced above in Figures R3 and R4, and Table R4) and data not shown). EOM results presented in the revised manuscript (Figs. 4g-l, Table 1, and Supplementary Fig. 6e-g (reproduced above in Figure R3, Table R4, and Figure R4)) are based on curves measured at the lowest concentrations of each diluted antibody series since all low concentration samples returned low χ^2 values (< 2) (Supplementary Fig. 6e and Table 1 in the revised manuscript (reproduced above in Figure R4e and Table R4)). All other SAXS data analyses (Guinier analysis, $P(R)$ analysis, and Kratky plots) performed in the previous and revised manuscript are based on the curves measured at the highest concentrations of each diluted antibody series. To clarify the point raised by the reviewer, a statement "SAXS data obtained from samples with the highest concentrations were used for Guinier analysis, $P(R)$ analysis, and Kratky plots, whereas data obtained from samples with the lowest concentrations were used for the Ensemble Optimization Method (EOM)" is added into the revised manuscript at line 589.

Main remarks concern the results in the table presented in Fig. 4G:

- How is $R_g(\text{\AA})$ STDEV calculated? A value as high as 8% makes the results suspect.

R_g STDEV values in Fig. 4G of the previous manuscript are calculated by the command-line AUTORG program (version 3.2, ATSAS 2.8.4 (r10552), <https://www.embl-hamburg.de/biosaxs/manuals/autorg.html>)⁴. The AUTORG calculation was performed automatically without any adjustment to the default settings.

When we were reanalyzing the SAXS data, we noticed that the R_g STDEV values returned, for the same SAXS data (same data files and same range of points), by the command-line AUTORG and the “Autorg” (“Radius of Gyration”) function of the program PRIMUSQT (the ATSAS program suite, Version 2.8.4)⁵ are different. They differ in a way that the command-line AUTORG program returns much larger R_g STDEV values in some samples. For instance, the command-line AUTORG and the PRIMUSQT autorg programs, respectively, return 69.3 ± 4.9 (or 7%) and 69.3 ± 0.98 (or 2%) for IgG3 antibodies (See below in Figures R6a-b, G3.dat with Guinier points 9~20); 54.0 ± 4.4 (or 8%) and 54.0 ± 0.27 (or 1%) for V11(H1) antibodies (See below in Figures R6a, c, V11H1.dat with Guinier points 16~33). It appears that, for the command-line AUTORG, adjusting the minimum acceptable Guinier interval length in points to 15 can reduce the R_g STDEV values (see below in Figure R6d). To simplify our analysis, we switched to the PRIMUSQT autorg program for Guinier analysis in the revised manuscript. As shown in Table 1 in the revised manuscript (reproduced below as Table R3), all R_g STDEV values are less than 2%. To clarify the point raised by the reviewer, the method is revised to “The radius of gyration (R_g) and I₀ were calculated by the “Autorg” (“Radius of Gyration”) function of the program PRIMUSQT (the ATSAS program suite, Version 2.8.4)⁵” in the revised manuscript at line 591.

Given the pattern of SAXS scattering intensity plots (Supplementary Fig. 6b), consistent results for Guinier plots (Supplementary Fig. 6c), Kratky plots, P(R) and EOM analyses (Fig. 4, Table 1,

Supplementary Fig. 6, and data not shown) at different concentrations, consistent R_g and $I(0)$ values returned by Guinier and $P(R)$ analyses (Supplementary Table 3c), as well as their consistency with the study by Tian et al.,³ we believe that our SAXS data are suitable to evaluate the flexibility of our antibodies.

```

a
fubin@ubuntu:~/Desktop/Shared/H/EOM/SAXS_reanalysis$ autorg -v
autorg 3.2, ATSAS 2.8.4 (r10552)
Copyright (c) ATSAS Team, EMBL, Hamburg Outstation, 2007-2018
fubin@ubuntu:~/Desktop/Shared/H/EOM/SAXS_reanalysis$ autorg G3.dat V11H1.dat
Rg stdev I(0) Guinier points Quality File
69.25 7% 136 9 - 20 ( 12) 82% G3.dat
53.96 8% 161 16 - 33 ( 18) 82% V11H1.dat
fubin@ubuntu:~/Desktop/Shared/H/EOM/SAXS_reanalysis$ autorg G3.dat
Rg = 69.3 +/- 4.9 (7%)
I(0) = 136 +/- 0.83
Points 9 to 20 (12 total)
Quality: 82%
fubin@ubuntu:~/Desktop/Shared/H/EOM/SAXS_reanalysis$ autorg V11H1.dat
Rg = 54 +/- 4.4 (8%)
I(0) = 161 +/- 0.5
Points 16 to 33 (18 total)
Quality: 82%
fubin@ubuntu:~/Desktop/Shared/H/EOM/SAXS_reanalysis$

```

```

d
fubin@ubuntu:~/Desktop/Shared/H/EOM/SAXS_reanalysis$ autorg --mininterval 15 G3.dat
V11H1.dat
Rg stdev I(0) Guinier points Quality File
70.88 2% 139 1 - 19 ( 19) 99% G3.dat
54.18 1% 162 16 - 32 ( 17) 81% V11H1.dat

```

Figure R6 (Data not shown in the revised manuscript). Rg STDEV values returned by the command-line AUTORG (version 3.2, ATSAS 2.8.4 (r10552))⁴ and the “Autorg” function of the program PRIMUSQT (the ATSAS program suite, Version 2.8.4)⁵ are different: (a) The command-line AUTORG returns 69.3 ± 4.9 (or 7%) for G3 and 54.0 ± 4.4 (or 8%) for V11(H1); (b-c) The “Autorg” in PRIMUSQT returns 69.3 ± 0.98 (or 2%) for G3 and 54.0 ± 0.21 (or 1%) for V11(H1). (d) The command-line

AUTORG returns smaller STDEV values when mininterval (the minimum acceptable Guinier interval length in points) is set to;

- How is Mr calculated from the Porod volume? To obtain a more reliable determination of Mr the authors have to use the module “Molecular Weight” available in PRIMUSQT which combines four concentration independent MM estimators.

*We thank the reviewer for suggesting the methods for calculating Mr values. In our study, Mr is estimated based on Porod volume (V_P) of analyzed antibodies and a standard protein (BSA, 66KD) using the following equation: $Mr(\text{sample}) = V_P(\text{sample}) / V_P(\text{BSA}) \times Mr(\text{BSA})$. $V_P(\text{BSA})$ is 114 nm^3 in our SAXS study. This method is based on the relationship between Mr, the scattering invariant (Q_i) and V_P , which was previously described by Porod and Debye (Porod, G. (1951). *Kolloid-Z.* 124, 83–114; Debye, P et al., (1957). *J. Appl. Phys.* 28, 679–683.) and recently summarized by Trehwella et al. (Trehwella et al., *Acta Cryst.* (2017). D73, 710). This method is part of the module “Molecular Weight” available in PRIMUSQT (suggested by the reviewer).*

To clarify the point raised by the review, the relevant method statement is revised to “Molecular mass (Mr) values were calculated based V_P values of analyzed antibodies and the BSA standard protein (66 KD), which has a V_P value of 114 nm^3 in our SAXS analysis, using the following equation: $Mr(\text{sample}) = V_P(\text{sample}) / V_P(\text{BSA}) \times Mr(\text{BSA})$.” in the revised manuscript at line 597.

- The values of Mr extracted from the SAXS data are higher than the values calculated from sequence in almost all cases. The difference could be ascribed to the presence of larger objects in very small proportion. The contribution to the SAXS curve of these larger objects could be also responsible for the surprisingly high values of Dmax in the case of G3, V11(H3), G2(H3) (see comment below).

Since it is technically challenging to completely remove “larger objects in very small proportion” that are not detectable, we could not completely rule out the possible contribution of such “larger objects in very small proportion” to our results. However, we reason that such contribution is not likely to have a significant impact on the major conclusions of our SAXS study regarding flexibility since our samples are processed in the same way and have no detectable aggregates as evaluated by SEC (Supplementary Fig. 6a, reproduced above in Figure R4a). We also reason that the difference between estimated M_r based sequences and SAXS is more likely due to the following two factors:

1) Antibodies have glycosylations, which is not considered in the M_r calculated based on sequences (a 3.2 KD glycan is present in the crystal structure of human IgG1 antibody (PDB: 1hzh) at the N297 site, and this glycosylation site is conserved in all human IgGs). At the same time, the glycosylation form is not fixed and therefore not considered in the calculation of M_r ;

*2) As suggested by Tian et al. (Tian et al., *Lucrij* 2, 9-18 (2015))³ and Rambo et al. (Rambo et al., *Biopolymers* 95, 559-571 (2011))⁶, flexible domains occupy volumetric space in solution and lead to overestimation of V_P for the calculation of M_r values. Interestingly, it appears that M_r values calculated based V_P for IgG2 and V11(H2) — two samples with the least flexibility [as assessed by TR-FRET (Figs. 5c-e in the revised manuscript) and by SAXS with both Kratky plots (Figs. 4a, c, e) and EOM modeling (Figs. 4g-l, and Table 1 in the revised manuscript, reproduced above in Figure R3 and Table R4)] — also have the smallest deviations from the M_r values calculated based on sequences.*

The relatively high values of D_{max} of G3, V11(H3), G2(H3) are more likely due to their IgG3 CH1-hinge (62 a.a.), which is much longer than the CH1-hinges of other analyzed antibodies (12~15 a.a.) (Supplementary Table 2 in the revised manuscript, reproduced above as Table

R2), as well as their extended conformations suggested by the Kratky plots and $Pr(R)$ analysis (Figs. 4a-f in the revised manuscript, reproduced above as Figures R3a-f), which is further supported by EOM modeling (Supplementary Fig. 6g in the revised manuscript, reproduced above as Figure R4g).

In brief, the authors should present in a revised manuscript a more complete Table following recommendations recently reported in Trehwella et al. (2017 publication guidelines for structural modelling of small-angle scattering data from biomolecules in solution: an update, J. Trehwella et al., *Acta Cryst.* (2017). D73, 710).

*Again, we thank the reviewer for the in-depth review and very helpful comments on our SAXS data. As suggested by the reviewer, we have redone the SAXS data analysis and presented updated and new results in the revised manuscript (the updated Figs. 4a-f, Supplementary Figs. 6c-e; the updated Table 1 (to replace the original Fig. 4G in the previous manuscript); new Figs. 4g-l, supplementary Figs. 6f-g, Supplementary Table 3), including two more complete tables (Table 1 and Supplementary Table 3 in the revised manuscript, which are reproduced above as Table R4 and R3, respectively) as recommended in Trehwella et al. (2017 publication guidelines for structural modeling of small-angle scattering data from biomolecules in solution: an update, J. Trehwella et al., *Acta Cryst.* (2017). D73, 710), and suggested by the reviewer.*

2. SAXS data interpretation

The authors have to be careful while using the word “flexibility”, as it is nicely explained in “Comparisons of the ability of human IgG3 Hinge Mutants, IgM, IgE, and IgA2, to form small Immune complexes: A Role for Flexibility and Geometry” (K. H. Roux and al., *J. Immunol.*

161(1998)4083).

The only definitive conclusion that can be extracted from the SAXS data without modelling is that G3, V11(H3) and G2(H3) are much more extended than the other antibodies. Concluding on flexibility is much more complicated.

It is unfortunate that the authors seem unaware of recent SAXS studies of antibodies IgG (Tian et al., J. Pharm. Sci. 103(2014)1701, Rayner et al., JBC 290(2015)8420) and especially the excellent work "In-depth analysis of subclass-specific conformational preferences of IgG antibodies", X. Tian, B. Vestergaard, M. Thorolfsson, Z. Yang, H. B. Rasmussen and A. E. Langkilde, IUCrJ 2(2015)9. Due to the flexible linkers connecting the Fab and Fc domains, IgG antibodies adopt in solution several conformations. As explained in the above article, it is thus absolutely necessary to describe the SAXS data in terms of structural ensembles for example by using the program EOM which "quantifies" the flexibility. The main result is that IgG1 and IgG2 antibodies are both flexible in solution but adopt different types of conformations: "IgG2 adapt to fewer overall conformations in solution, while IgG1 reveals a continuum of conformations around the preferred intermediate Y-shape" as nicely represented by the fig.3 of this article. It should be noted that these results are not in agreement with the FRET data shown in the manuscript of Liu et al.

In conclusion, authors must perform new analysis of their data (or preferably of improved data devoid of any contributions due to larger objects) by using such an approach.

I look forward to seeing the results for G3, V11(H3) and G2(H3) because the very high value obtained for D_{max} leads to the suspicion that IgG3 adopts always very extended conformations, and thus are not very flexible. This would disagree with the FRET data! It is even very strange that the D_{max} value (of the order of 250-270 Å) is higher than the most extended conformations described by Tian et al. (of the order of 200 Å). I therefore believe that a small amount of larger objects could be present in the samples.

We agree with the reviewer that G3, V11(H3) and G2(H3) are much more extended than the other antibodies, and that extended conformations are different from flexibility. We also agree that while dimensionless Kratky plots can provide evidence for flexibility (as reviewed in J. Trehwella et al., Acta Cryst. (2017). D73, 710), it will be helpful to use the program EOM to “quantify” the flexibility (see below and the revised manuscript for the results) in addition to dimensionless Kratky plots.

We thank the reviewer for citing the study by Tian et al., (Tian et al., IUCrJ. 2015 Jan 1;2(Pt 1):9-18.)³, which is cited in the revised manuscript (Reference No. 32 in the revised manuscript). Initially, this study was not cited because the focus of our SAXS study — the flexibility difference between antibodies with human IgG3 and IgG1/2 CH1-hinge sequences — was not investigated by Tian et al.

At the same time, we disagree with the reviewer regarding the point “It should be noted that these results are not in agreement with the FRET data shown in the manuscript of Liu et al.” Our TR-FRET data show that IgG2 is the most rigid among IgG1, 2 and 3. In Tian et al’s study cited by the reviewer, the authors state “IgG2 has the narrowest distribution profile (of R_g and D_{max}) of the three IgGs (IgG1, 2 and 4), reflecting a relatively high rigidity with a fairly stable particle size” (Beginning at the 5th line counting from the bottom at the low right corner on page 13, Tian et al., IUCrJ. 2015 Jan 1;2(Pt 1):9-18., [REDACTED], which is completely consistent with the conclusion of our TR-FRET study .

[REDACTED].

As suggested by the reviewer, we performed EOM analysis of our SAXS data and came into the same conclusion that human IgG3 is the most flexible among human IgG1, 2 and 3 due to its

CH1-hinge. Furthermore, EOM analysis also suggests that human IgG2 is the least flexible among human IgG1, 2 and 3 due to its CH1-hinge, a notion that is supported by our TR-FRET study but not appreciated in our previously SAXS data analysis. Specifically, we found that consistent with the study by Tian et al., the selected models of human IgG2 have narrower distribution profiles for R_g and D_{max} than those of human IgG1 (compare Figs. 2g and 2h in Tian et al. with Figs 4h and 4g of the revised manuscript), suggesting that human IgG2 is less flexible than human IgG1. We also found that the selected models of human IgG2 and 3 have the narrowest and broadest distribution profile, respectively (Figs 4g and 4h in the revised manuscript, reproduced above in Figure R3). The ranking of R_{flex} and R_σ values, a quantitative measure of flexibility, is IgG3 > IgG1 > IgG2 (Table 1 in the revised manuscript, reproduced above as Table R4). Importantly, the same trend was observed in V11 variants: V11(H3) > V11(H1) > V11(H2), suggesting that CH1-hinge sequences are responsible for these differences. Furthermore, analysis of G2(H3) and G3(H2) showed that the distribution profiles of R_g and D_{max} (Figs. 4k-l and Supplementary Fig. 6f in the revised manuscript, reproduced above in Figures R3 and R4), R_{flex} and R_σ rankings (Table 1 in the revised manuscript, reproduced above as Table R4) of human IgG2 and IgG3 antibodies could be switched along with their CH1-hinges. These results are not only consistent with the IgG1 and IgG2 part of the study by Tian et al. [REDACTED], but also with our TR-FRET studies, further supporting our conclusion that human IgG3 has the most flexible CH1-hinge whereas IgG2 has the most rigid CH1-hinge among human IgG1, 2 and 3. A paragraph describing these results is added into the revised manuscript at line 284 as the following:

“ The ensemble optimization method (EOM) for SAXS data has been developed to evaluate flexibility quantitatively^{3, 7}. The application of this method to our SAXS data yields high-quality fits between the optimized ensemble and the experimental data (Supplementary Fig. 6e). It appears that the selected models of human IgG2 have the narrowest distribution profiles of R_g (Fig. 4g)

and D_{max} (Figs. 4h and S6F) among those of human IgG1, 2 and 3, consistent with previous SAXS study of human IgG1 and 2 antibodies³. At the same time, the selected models of human IgG3 show the broadest profiles. These results suggest that human IgG2 and 3 are the least and most flexible among human IgG1, 2 and 3, respectively. This notion is further supported by the ranking of R_{flex} and R_{σ} values, quantitative measures of flexibility: IgG3 > IgG1 > IgG2 (Table 1). Importantly, V11 variants with human IgG1, 2 and 3 CH1-hinges have similar distribution profiles of R_g and D_{max} as human IgG1, 2 and 3 antibodies, respectively (Figs. 4i-j, Supplementary Fig. 6f), as well as R_{flex} and R_{σ} rankings: V11(H3) > V11(H1) > V11(H2) (Table 1). Furthermore, analysis of G2(H3) and G3(H2) showed that the distribution profiles of R_g and D_{max} (Figs. 4k-l, Supplementary Fig. 6f), R_{flex} and R_{σ} rankings (Table 1) of human IgG2 and IgG3 antibodies could be switched along with their CH1-hinges. Models generated by EOM analysis suggest that the long IgG3 hinge is highly flexible and can support various conformations of IgG3, V11(H3) and G2(H3) antibodies (Supplementary Fig. 6g). Overall, our SAXS study suggests that the flexibility of human IgG1, 2 and 3 antibodies is primarily determined by their CH1-hinges and that among them human IgG2 and 3 has the least and most flexible CH1-hinges, respectively.

The observation that the D_{max} value of human IgG3 (of the order of ~ 270 Å) is higher than the most extended conformations described by Tian et al. (of the order of 200 Å) might not be that “strange” given that: 1) human IgG3 has a much longer hinge (62 a.a., Supplementary Table 2, reproduced above as Table R2) as compared to human IgG1, 2 and 4 (12~15 a.a.), which are what Tian et al. studied (X. Tian, B. Vestergaard, M. Thorolfsson, Z. Yang, H. B.Rasmussen and A. E. Langkilde, IUCrJ 2(2015)9)³; 2) human IgG3 CH1-hinge has more extended conformations (Figs. 4a, c, e in the revised manuscript, reproduced above in Figure R3) and is more flexible than the CH1-hinge regions of human IgG1 and 2, as suggested by both SAXS and TR-FRET studies. The predicted larger size of human IgG3 antibodies based on these two points is also visualized

and supported by EOM modelling (Supplementary Fig. 6g in the revised manuscript, reproduced above in Figure R4.).

Reviewer #4 (Remarks to the Author):

This manuscript addresses an unresolved issue relating to the structure/function mechanisms that contribute to immune stimulation mediated by agonist human IgG antibodies. The authors have previously shown that human IgG1 and IgG2 agonistic anti-CD40 antibodies depend on FcγR binding, with FcγRIIB binding demonstrating improved agonism. However, others have shown that human IgG2 agonism does not depend on Fc-FcγR interactions, but rather on hinge conformation. This becomes more complicated as there is evidence that the impact of IgG2 hinge conformation may not apply to all anti-CD40 antibody clones and may not apply in FcγR-humanized mice. Therefore, the authors examine if IgG hinge and FcR-interactions impact anti-CD40 and anti-DR5 antibody agonism using FcγR-humanized mice. IgG hinge biophysical flexibility was investigated using different natural and engineered human IgG Fc's and CH1-hinges to determine their influence on agonistic function.

The authors found that different IgGs induce different agonistic responses and that IgG hinge and Fc domains together regulate agonism through specific mechanisms. They show that IgG2 antibodies are highly agonistic whilst IgG3 antibodies are inactive. This activity was primarily mediated by the biophysical flexibility of CH1-hinges, although Fc-FcγR binding was still required for optimal agonistic outcomes. The authors compare the efficacy of their anti-CD40 antibodies to unmodified antibody in murine models of colon cancer and melanoma and demonstrate that their modified anti-mouse CD40V11(H2) provides significantly improved anti-tumour efficacy (though the figures were difficult to read). They also show that the effects of

hinge rigidity/flexibility on agonism are not limited to anti-CD40 antibodies as they could be reproduced in agonistic anti-DR5 antibodies.

Overall, this is a solid and very interesting paper, worthy of publication. The authors provide a resolution for the apparently conflicting data published by themselves and others. The novelty of their findings lies in demonstrating the importance of both hinge rigidity and selective FcγR (i.e. FcγRIIB) binding in antibody agonistic function. These findings may indeed represent improved strategies to improve antibody agonism for clinical application.

We are pleased that the reviewer found our findings interesting. We thank the reviewer for pointing out the issues in the manuscript and providing very helpful comments.

Major concerns

- Animal ethics approval numbers should be provided.
- In the figure legends please state the number of mice used in each experimental group for all figures as it is sometimes difficult to see.

Animal ethics approval numbers are now provided in the “Methods” section of the revised manuscript, which reads “All animal experiments were performed in compliance with institutional guidelines and had been approved by SJTUSM Institutional Animal Care and Use Committee (Protocol Registry Number: A-2015-014),” at line 484.

As suggested by the reviewer, the number of mice used in each experimental group is now stated in all figure legends in the revised manuscript.

Re Figure 1: Please state that the N297A mutation abrogates Fc-FcγR binding on the figure or in the figure legend.

As suggested by the reviewer, “(the N297A mutation abrogates Fc-FcγR binding)” is now added into the legend of Figure 1 at line 928.

Re Supp Fig 1:

a. When were the samples collected?

b. There also seem to be missing data sets, e.g. from control untreated mice, and mice treated with PIGF-2123-144-aCD40.

Based on the questions, we interpreted that these comments are for Figure 1 (instead of Supplementary Fig. 1).

a) As shown in Fig. 1a, samples were collected 6 days after immunization to quantify OVA-specific CD8⁺ T cell responses. To clarify the point raised by the reviewer, the Figure legends are revised to “Splenocytes were harvested to quantify OVA-specific CD8⁺ T cells on day 6.” at line 920 in the revised manuscript.

b) We agree with the reviewer that it will be good to have information on the basal OT-I cell levels in control IgG-treated FcγR-deficient (FcγRα^{-/-}), which has been analyzed in Fig. 1g in the

manuscript (reproduced below as Figure R8). In this experiment, we found that control IgG and human IgG2 anti-CD40 antibody-treated *FcγR*-deficient (*FcγRα*^{-/-}) mice have the same OT-I cell levels, supporting the conclusion that human IgG2 anti-CD40 antibodies have no detectable agonistic activities in *FcγR*-deficient (*FcγRα*^{-/-}) mice. With the information in Fig. 1g, we reason that control IgG-treated *FcγR*-deficient (*FcγRα*^{-/-}) mice are not absolutely required in Figs. 1b-c, where the impact of *FcγR*-deficiency on anti-CD40 antibody agonism is investigated by directly comparing the agonistic activities of anti-CD40 antibodies in *FcγR*-humanized (*hFCGR*^{Tg}) and *FcγR*-deficient (*FcγRα*^{-/-}) mice.

Figure R8. (Fig. 1g in the revised manuscript) Quantification of OT-I cells as the percentage of OT-I cells among CD8⁺ T cells in mice of indicated genotypes (*FcγR*-deficient (*FcγRα*^{-/-}, 5 mice per group), *FcγRIIB*-deficient (*FcγR2b*^{-/-}, 5 mice per group) or humanized (*FcγR2b*^{-/-} *hFCGR2B*^{Tg}, 4 mice per group)) treated and analyzed as in (a) together with 10 μg of indicated control or IgG2 anti-mCD40 antibodies. Each symbol represents

an individual mouse. Bars represent mean ± SEM. $p \leq 0.001$, **** $p \leq 0.0001$; unpaired two-tailed *t*-test.

PIGF-2123-144-aCD40 mentioned by the reviewer is an anti-CD40 variant designed to increase extracellular matrix binding and to prolong tissue retention of anti-CD40 antibodies, which has been suggested to improve antibody half-life and activities while reducing the toxicity (Mol Cancer Ther. 2018 Nov;17(11):2399-2411). *FcγR*-independent interactions between anti-CD40 antibodies and the tissue environment is an important aspect of anti-CD40 antibody optimization, which is now included in the discussion in the revised manuscript at line 422. At the same time, PIGF-2123-144-aCD40 focuses on adding new properties to anti-CD40 antibodies, rather than

optimizing IgG constant domain-intrinsic properties — the focus of the current study. Therefore, we think it is not essential to study PIGF-2123-144 antibodies in our study.

Re: Figure 2: Please clarify what anti-hCD4021.4.1 in the legend.

As suggested by the reviewer, a brief description “Clones 21.4.1 and 3.1.1 have been described in Patent No.:US 7,338,660; Clone 21.4.1 in the IgG2 form is also known as CP-870,893” has been added into the legend at line 946 of the revised manuscript.

Re Figure 3:

- There is a typo in the legend re (A-C) and (A-B)

“(A-C)” is now deleted in the updated legends to improve the clarity.

- It is not clear what day 0 is. Is it the day of tumour cell inoculation, or when palpable tumours could be detected?

Day 0 is the day when mice with palpable tumours receive their first treatment of control or anti-CD40 antibodies. As the reviewer pointed out, the original Figure legends are not clear about the definition of day 0. The relevant Figure legends are revised to “After tumour cells were subcutaneously inoculated and established in FcγR-humanized mice, mice were treated i.p twice on day 0 (the day when mice with palpable tumours receive their first treatment) and 3

with 31.6 µg/mouse of control or anti-CD40 antibodies of indicated constant domains ...” at line 956.

- How big were the tumours when treatment commenced - you cannot tell from the graphs. My main concern is that differing tumour sizes could respond differently to anti-CD40 antibody and it is difficult to tell on the graph if size impact response to the different antibodies. For example, the control IgG treated tumour sizes look like they might be slightly larger than the ones given anti-CD40:V11(H2).
- Figures 3B and C make it very difficult to see all treatment groups. Please make the figures bigger. It would also be helpful state what N2974 is to remind the reader.

When the treatment starts, tumour sizes are 0.002 ~ 0.05 cm³ for MC38, and 0.001 ~ 0.07 cm³ for MO4 (Individual values are provided in the source data file “SourceData.xlsx”). As shown in the updated Figs. 3b and 3c (reproduced below as Figure R9), and the source data, all groups have the same distribution, average and variations for tumour sizes when the treatment starts (day 0).

We have enlarged and reformatted Figs. 3b and 3c in the revised manuscript (reproduced below as Figure R9) to improve the readability. As suggested by the reviewer, “(the N297A mutation abrogates Fc-FcγR binding)” is added into the legend of Figure 3 at line 957 in the revised manuscript.

Figure R9. (Figs. 3b and 3c in the revised manuscript). MC38 (b) and MO4 (c) tumour volumes in Fc γ R-humanized mice following treatment with control or anti-mCD40 antibodies of indicated constant domains. After tumour cells were subcutaneously inoculated and established in Fc γ R-humanized mice, mice were treated *i.p* twice on day 0 (the day when mice with palpable tumours receive their first treatment) and 3 with 31.6 μ g/mouse of control or anti-CD40 antibodies of indicated constant domains (the N297A mutation abrogates Fc-Fc γ R binding), and monitored for tumour growth. For mice inoculated with MO4 tumour cells in (c), each treatment also included 2 μ g/mouse of DEC-OVA. Shown are tumour growth curves. Numbers of mice: (b) 7–8 mice per group; (c) 7 mice per group except 6 mice for

amCD40:G2(N297A). Mean \pm SEM is presented. * $p \leq 0.05$, ** $p \leq 0.01$, *** $p \leq 0.001$, **** $p \leq 0.0001$, two-way ANOVA with Holm-Sidak's post hoc (**b-c**) were used for group comparison.

Re Figures 4, 5, S6 and S7: These are very interesting data, however the methods used are out of my field of expertise.

- The legend for figure 5 needs correcting – e.g. A diagram showing anti-mCD40 antibody TR-FRET and B) A diagram of the model showing that hinge flexibility of human IgG anti-mCD40 antibodies inversely correlates with TR-FRET signal levels

We thank the reviewer for pointing out these important issues in writing, which have been corrected in the revised manuscript at line 971.

- For Figure 6 it would be helpful to say what G2(Gs)3 is.

As suggested by the reviewer, “(G2(GS)₃ is a human IgG2 constant domain variant with a “GSGSGS” insertion in the hinge (Supplementary Table 2))” is now added into the legend of Fig. 6 in the revised manuscript at line 988, and a new supplementary table summarizing all applied hinge sequences is included in the revised manuscript as Supplementary Table 2 (reproduced below as Table R2).

Table R2 (Supplementary Table 2 in the revised manuscript). Hinge and Fc sequences of various human IgG constant domains.

Constant domains	C-terminal of CH1	Upper hinge	Middle hinge	Hinge length	Fc
G1	NTKVDKRV	EPKSCDKTHT	CPPCP	15	G1
G2	NTKVDKTV	ERK	CCVECPPCP	12	G2
G2(GS) ₃	NTKVDKTV	ERKGSGSGS	CCVECPPCP	18	G2
G3(H2)	NTKVDKTV	ERK	CCVECPPCP	12	G3
G3	NTKVDKRV	ELKTPLGDTTHT	CPRCP(EPKSCDTPPPCPRCP) ₃	62	G3
G2(H3)	NTKVDKRV	ELKTPLGDTTHT	CPRCP(EPKSCDTPPPCPRCP) ₃	62	G2
G4	NTKVDKRV	ESKYGPP	CPSCP	12	G4
V11(H1)	NTKVDKRV	EPKSCDKTHT	CPPCP	15	V11
V11(H2)	NTKVDKTV	ERK	CCVECPPCP	12	V11
V11(H3)	NTKVDKRV	ELKTPLGDTTHT	CPRCP(EPKSCDTPPPCPRCP) ₃	62	V11

Re Figure 7:

- What do the crosses on Fig 7D symbolize?

Grosses symbolize mortality. A note “(crosses symbolize mortality)” is now added into the legend of Fig 7 in the revised manuscript at line 1003.

- How can the difference between control IgG and the others be statistically significant, there are only 2 mice in one group?

We are sorry that the original description for the number of mice used in the experiment is not clear. There are 4~7 mice per group (mortality is symbolized by crosses). This problem is now

fixed in the revised manuscript by adding a note “(crosses symbolize mortality)” into the legend of Fig. 7 in the revised manuscript at line 1003.

Re figure S3: I found the figure confusing, I thought G2 bound FcγR2B?

Yes, it has been established that human IgG2 binds to human FcγR2B (Bruhns P, et al. *Blood* 113, 3716-3725 (2009))⁸. We have also confirmed it by surface plasmon resonance analysis (New Supplementary Fig. 3b in the revised manuscript, reproduced below as Figure R10). However, the affinity between monomeric IgG2 and FcγR2B is too low to be detected in our ELISA. To clarify this confusion, a point “The binding between human IgG2 antibodies and FcγR2B, although undetectable in ELISA (Supplementary Fig. 3a) likely due to its low affinity⁸, was confirmed by surface plasmon resonance (Supplementary Fig. 3b)” is added into the revised manuscript at line 153. The “Surface plasmon resonance (SPR)” method is added into the revised manuscript at line 667.

Figure R10 (Supplementary Fig. 3b in the revised manuscript) SPR analysis of the binding of human IgG1 and 2 antibodies to human FcγR2B. Presented are real-time sensorgrams with affinity constants (KD).

Re typos and English:

1. Line 30 – should read ‘...human IgGs requiring Fc-FcγR binding...
2. Line 158 should say ‘investigated’
3. Line 186: ‘antibodies having comparable mCD40 and human FcγR binding profiles with V11(H1) antibodies (Figs. S1B and S3B). These results demonstrate that human IgG3 CH1-hinge deprives V11 Fc of its strong agonistic potency.’
4. Line 212 Title: ‘The impact of human IgG CH1-hinge and Fc domains on anti-CD40 antibody immunostimulatory activities can be translated into antitumor activities’
5. Line 330: ‘Importantly, IgG2(GS)3 anti-mCD40 antibodies displayed clearly reduced immunostimulatory activities as compared to matched control IgG2 antibodies (Fig. 6B)’
6. Line 361: ‘This activity was abrogated either by co-culturing with FcγR-deficient cells or by the addition of human FcγRIIB blocking antibody 2B6 (Fig. 7B), suggesting that human FcγRIIB engagement is required’
7. Line 371: ‘In contrast, mice treated with V11(H3) anti-DR5 antibodies were mostly protected.’
8. Line 378: ‘Interestingly, despite all human IgGs requiring Fc-FcγR engagement....’
9. Line 382: ‘Our study demonstrates that in the human FcγR-expressing background, the human IgG CH1-hinge is directly implicated in regulating antibody agonistic function, and has a dominant contribution to the observed divergent agonistic potency of natural human IgG constant domains.’
10. Line 392: ‘we speculate that both IgG CH1-hinge and Fc contribute to antibody-mediated in

vivo clustering and crosslinking of the targeted receptors, a proposed mechanism explaining how...'

11. Line 532: 'spleens were harvested'

12. Line 587: 'MO4 is an OVA-expressing B16F10 please state this melanoma cell line previously described'

We thank the reviewer very much for pointing out these typos and English issues. All these issues have been fixed in the revised manuscript.

References

1. *Richman LP, Vonderheide RH. Role of crosslinking for agonistic CD40 monoclonal antibodies as immune therapy of cancer. Cancer immunology research 2, 19-26 (2014).*
2. *Bedian V, Gladue RP, Corvalan J, Jia X, Feng X. Methods of treating cancer and enhancing immune responses with antibodies that bind CD40 Patent Patent No.:US 7,338,660 (2008).*
3. *Tian X, Vestergaard B, Thorolfsson M, Yang Z, Rasmussen HB, Langkilde AE. In-depth analysis of subclass-specific conformational preferences of IgG antibodies. Iucrj 2, 9-18 (2015).*
4. *Petoukhov MV, Svergun DI. Analysis of X-ray and neutron scattering from biomacromolecular solutions. Curr Opin Struct Biol 17, 562-571 (2007).*
5. *Konarev PV, Volkov VV, Sokolova AV, Koch MHJ, Svergun DI. PRIMUS: a Windows PC-based system for small-angle scattering data analysis. Journal of Applied Crystallography 36, 1277-1282 (2003).*
6. *Rambo RP, Tainer JA. Characterizing flexible and intrinsically unstructured biological macromolecules by SAS using the Porod-Debye law. Biopolymers 95, 559-571 (2011).*

7. *Tria G, Mertens HD, Kachala M, Svergun DI. Advanced ensemble modelling of flexible macromolecules using X-ray solution scattering. Iucrj 2, 207-217 (2015).*
8. *Bruhns P, et al. Specificity and affinity of human Fcγ receptors and their polymorphic variants for human IgG subclasses. Blood 113, 3716-3725 (2009).*

Reviewers' comments:

Reviewer #2 (Remarks to the Author):

The authors provide a revised manuscript in which they have addressed all issues raised by the reviewers. The manuscript is acceptable for publication.

Reviewer #3 (Remarks to the Author):

The authors have redone all SAXS data analysis. Consequently, they have considerably improved their manuscript which did become worthy of publication. Nevertheless, I have noticed two minor points that must be corrected.

Concerning the Guinier analysis and the EOM analysis:

In Figure 6c the authors must provide the R_g values for the different concentrations. The use of the value obtained at the highest concentration makes sense only if these values are equal. Otherwise, they must provide the R_g value extrapolated at zero concentration. Concerning EOM, it is obvious that the curve measured at the lowest concentration gives the lowest χ^2 value since the error bars are very large!! But the authors have to use the curve with the best statistic, either the curve at the highest concentration, or the curve extrapolated at zero concentration.

Concerning the molecular mass determination:

Why do the authors use only a part of the module "Molecular Weight" available in PRIMUSQT? They have to use the full module and to give the values determined using the bayesian inference approach which combines four concentration independent MM estimators. It should be noted that the use of BSA is dangerous because a BSA solution contains always a non-negligible part of dimers.

Reviewer #4 (Remarks to the Author):

Overall I was happy with the responses to my questions and comments. I also thought that the responses to other reviewers made this a much more meaningful paper.

The figure legend for 3b and c still requires greater clarity.

It is not clear what the authors are referring to when they state 'mice were treated i.p twice on day 0 (the day when mice with palpable tumours receive their first treatment) and 3 with 31.6 $\mu\text{g}/\text{mouse}$ of control or anti-CD40 antibodies of indicated constant domains' - what does the '3' refer to?

Please note that whilst I did not request the information I think that Supplementary Table 1 should state exactly what the epitopes are rather than just refer to references.

Please note that the authors' point-by-point response is italicized.

We thank all reviewers for providing their comments, which are very helpful for our manuscript.

Reviewers' comments:

Reviewer #2 (Remarks to the Author):

The authors provide a revised manuscript in which they have addressed all issues raised by the reviewers. The manuscript is acceptable for publication.

Reviewer #3 (Remarks to the Author):

The authors have redone all SAXS data analysis. Consequently, they have considerably improved their manuscript which did become worthy of publication.

Nevertheless, I have noticed two minor points that must be corrected.

Concerning the Guinier analysis and the EOM analysis:

In Figure 6c the authors must provide the R_g values for the different concentrations. The use of the value obtained at the highest concentration makes sense only if these values are equal.

Otherwise, they must provide the R_g value extrapolated at zero concentration. Concerning EOM, it is obvious that the curve measured at the lowest concentration gives the lowest χ^2 value since the error bars are very large!! But the authors have to use the curve with the best statistic, either the curve at the highest concentration, or the curve extrapolated at zero concentration.

As suggested by the reviewer, R_g values (Mean \pm SD) for the different concentrations are provided in the updated Supplementary Fig. 6c in the revised manuscript (reproduced below as Fig. R1). As shown in this figure, The R_g values obtained at different concentrations are very close, and it is, therefore, justified to use the R_g values obtained from samples with the highest concentrations.

Figure R1 (reproduced from the updated Supplementary Fig. 6c in the revised manuscript). **Guinier plots ($\ln I(q)$ vs q^2) of indicated antibodies at different concentrations** colour-coded as in Supplementary Figure 6b (blue, high concentration; red, middle concentration; black, low concentration), with R_g values (Mean \pm SD) annotated.

As suggested by the reviewer, we have also performed EOM analysis using the curves at the highest concentrations and added the results into the revised manuscript, including the new Supplementary Figures 6h-k, Supplementary Table 4. Both low- and high- concentration EOM results are included in the revised manuscript (Low-concentration: Table 1, Figures 4g-l, Supplementary Figures 6e-g; high-concentration: Supplementary Figures 6h-k, Supplementary Table 4) and reproduced below as Table R1 and Figure R2. As shown in these figures and tables, despite that the EOM analysis of high-concentration curves returns higher χ^2 values, the results returned by the EOM analyses of high- and low- concentration curves are essentially the same. Regarding the flexibility levels of analyzed antibodies, the following ranking can be established based on the distribution profiles of R_g and D_{max} (Figures 4g-l, Supplementary

Figures 6i-j), and the ranking of R_{flex} and R_{δ} values (Table 1, Supplementary Table 4): (1) $IgG3 > IgG1 > IgG2$; (2) $V11(H3) > V11(H1) > V11(H2)$; (3) $G2(H3), G3 > G2, G3(H2)$). In the revised manuscript, the following statement is added into the result section at line 300 to describe these new results: “Further EOM analysis using high-concentration samples returned essentially the same results (Supplementary Figs. 6h-k, Supplementary Table 4).” The relevant methods are revised to “... data obtained from samples with both the highest and lowest concentrations were used for the Ensemble Optimization Method (EOM)” at line 591. Concentration information is now provided in the legends of Figure 4 and Supplementary Figure 6, where SAXS data are presented.

Table R1 . EOM results of low- and high-concentration antibodies samples (reproduced from Table 1 and Supplementary Table 4, respectively). (EOM setting: default parameters, 10,000 models in the initial ensemble, native-like models, constant subtraction allowed)

	G1	G2	G3	V11(H1)	V11(H2)	V11(H3)	G2(H3)	G3(H2)
EOM results of low-concentration samples (reproduced from Table 1)								
χ^2	1.190	1.180	1.118	1.476	0.933	1.311	1.102	1.661
No. of curves.	11	8	9	10	9	9	10	9
R_{flex} (random) (%)	~ 79.35 (~ 85.11)	~ 73.71 (~ 84.81)	~ 80.18 (~ 85.96)	~ 78.46 (~ 82.15)	~ 77.43 (~ 84.39)	~ 83.31 (~ 85.40)	~ 82.66 (~ 83.33)	~ 78.63 (~ 83.89)
R_{δ}	0.81	0.65	1.28	0.88	0.77	1.39	1.45	0.83
EOM results of high-concentration samples (reproduced from Supplementary Table 4)								
χ^2	3.525	2.602	3.220	5.798	1.966	5.228	2.652	9.833
No. of curves.	11	8	8	8	9	9	9	9
R_{flex} (random) (%)	~ 81.05 (~ 84.33)	~ 77.15 (~ 84.16)	~ 85.21 (~ 80.87)	~ 79.06 (~ 82.15)	~ 76.63 (~ 83.43)	~ 82.99 (~ 83.55)	~ 83.45 (~ 83.33)	~ 78.43 (~ 83.40)
R_{δ}	0.89	0.79	1.33	0.90	0.76	1.68	1.50	0.84

Low-concentration samples

Supplementary Fig. 6

Fig. 4

Supplementary Fig. 6 (continued)

High-concentration samples

Supplementary Fig. 6

Supplementary Fig. 6

Supplementary Fig. 6 (continued)

Figure R2 (Reproduced from Figure 4 and Supplementary Figure 6 of the revised manuscript), **the results returned by the EOM analyses of high- and low- concentration curves are essentially the same.** EOM results of low-concentration (left) and high-concentration curves of indicated antibodies: top, fits between the calculated scattering curve from the best ensemble (red line, selected by EOM) and the experimental data (black line) (Supplementary Figures 6e and 6h); middle, the distribution of R_g and D_{max} in the optimized ensembles generated in EOM analysis of the indicated antibodies (Figures 4g-l and Supplementary Figures 6i-j); bottom, models composing the best fitting ensemble generated by the EOM analysis of indicated antibodies (Supplementary Figures 6g and 6k).

Concerning the molecular mass determination:

Why do the authors use only a part of the module “Molecular Weight” available in PRIMUSQT?

They have to use the full module and to give the values determined using the bayesian inference approach which combines four concentration independent MM estimators. It should be noted that the use of BSA is dangerous because a BSA solution contains always a non-negligible part of dimers.

In our previous analysis, we estimated MW values based on Porod Volume (which is also the basis of several MW estimators used in the “Molecular Weight” module of PRIMUSQT) together with a reference protein, without realizing that the BSA is not a very good reference protein.

We thank the reviewer for pointing out that BSA is not a good reference protein for estimating M_r values and that the Bayesian Inference approach is a better approach (Sci Rep. 2018 May 8;8(1):7204). As suggested by the reviewer, we used the Bayesian Inference approach to analyze the M_r values of our samples and updated the results in Table 1 of the revised manuscript (reproduced below as Table R2). The analysis returns both the high probability M_r and Credibility Intervals. As shown in Table R2, all the calculated M_r credibility intervals of our

samples cover their Mr values calculated from their sequences; and the high probability Mr values are close to their Mr values calculated from sequences, especially for samples that are less flexible (IgG1, IgG2, et al.); we also consistently observed that antibodies that have larger deviations from the Mr values calculated from protein sequences are also more flexible (IgG3, V11(H3), G2(H3)). The relevant methods are revised to “Molecular mass (Mr) values were calculated using the Bayesian Inference approach¹ (the “Molecular Weight” module of the program PRIMUSQT²), at line 598 in the revised manuscript.

Table R2 (reproduced from Table 1). SAXS-derived parameters.

	G1	G2	G3	V11(H1)	V11(H2)	V11(H3)	G2(H3)	G3(H2)
Molecular Weight Analysis (KD)								
Mr by Bayesian Inference (%)	157 (69)	131 (46)	170 (29)	138 (42)	131 (51)	170 (33)	208 (37)	170 (53)
Credibility Interval (%)	142-177 (99)	116-151 (94)	127-177 (95)	127-151 (95)	121-151 (94)	134-177 (95)	99-264 (90)	151-195 (99)
Mr from sequence	145	144	155	145	145	155	155	144

Reviewer #4 (Remarks to the Author):

Overall I was happy with the responses to my questions and comments. I also thought that the responses to other reviewers made this a much more meaningful paper.

The figure legend for 3b and c still requires greater clarity.

It is not clear what the authors are referring to when they state 'mice were treated i.p twice on day 0 (the day when mice with palpable tumours receive their first treatment) and 3 with 31.6 µg/mouse of control or anti-CD40 antibodies of indicated constant domains' - what does the '3' refer to?

Thank the reviewer for pointing out this unclear statement, in which “3” was used to refer to day 3. This statement is now revised to “... mice were treated i.p twice on day 0 (the day when mice

with palpable tumours receive their first treatment) and day 3 with 31.6 µg/mouse of control or anti-CD40 antibodies of indicated constant domains” at line 960 in the revised manuscript.

Please note that whilst I did not request the information I think that Supplementary Table 1 should state exactly what the epitopes are rather than just refer to references.

Thank the reviewer for the suggestion. We have updated Supplementary Table 1 by adding specific epitope information that is available (reproduced below as Table R3).

Table R3 (reproduced from Supplementary Table 1). Binding epitopes of CD40 antibodies in relationship to CD40L binding sites.

Specificity	Clone	Binding epitope	References
Mouse CD40	1C10	Block CD40L binding	26
Human CD40	21.4.1	CRD1 of hCD40; do not block hCD40L binding	26, 27,
Human CD40	3.1.1	CRD2/3 of hCD40; block hCD40L binding	26, 27

CRD, cysteine-rich domain

References

1. Hajizadeh NR, Franke D, Jeffries CM, Svergun DI. Consensus Bayesian assessment of protein molecular mass from solution X-ray scattering data. *Sci Rep* **8**, 7204 (2018).
2. Konarev PV, Volkov VV, Sokolova AV, Koch MHJ, Svergun DI. PRIMUS: a Windows PC-based system for small-angle scattering data analysis. *Journal of Applied Crystallography* **36**, 1277-1282 (2003).

REVIEWERS' COMMENTS:

Reviewer #3 (Remarks to the Author):

I'm happy with the modifications made by the authors. The manuscript is acceptable for publication.
Dominique Durand

Please note that the authors' point-by-point response is italicized.

We thank all reviewers for providing their comments, which are very helpful for our manuscript.

REVIEWERS' COMMENTS:

Reviewer #3 (Remarks to the Author):

I'm happy with the modifications made by the authors. The manuscript is acceptable for publication.

Dominique Durand

We very much appreciate Dr Durand for valuable comments and suggestions regarding our SAXS data. These comments and suggestions have greatly helped our analysis of SAXS results and the manuscript.